# AGENT-REINFORCE: SEARCHING COMPUTE-OPTIMAL MULTI-LLM COLLABORATION GRAPH FOR TEST-TIME SCALING

## ABSTRACT

Test-Time Scaling (TTS) improves large language models (LLMs) by allocating additional computation during inference, typically through parallel, sequential, or hybrid scaling. However, prior studies often assume fixed collaboration architectures (e.g., topologies) and single-model usage, overlooking that optimal architectures and model combinations can vary across tasks. Therefore, we study the novel problem of *searching for compute-optimal model combinations and architectures in TTS under a fixed budget*. We formalize it as a multi-LLM collaboration graph, where nodes encode roles and LLM model assignments, and edges capture information flow. This problem is challenging because (i) the combinatorial search space is prohibitively large, and (ii) task-specific requirements demand tailored designs. To address these, we reformulate the problem as probabilistic graph optimization and, through pilot experiments, derive three empirical insights into TTS collaboration graphs. Guided by these insights, we propose Agent-REINFORCE, an LLM-agent-augmented framework that mirrors the REINFORCE pipeline by mapping *sampling–gradient–update* to *sampling–feedback–update*, where feedback serves as a textual gradient to update the probabilistic graph and efficiently search for optimal multi-LLM collaboration graphs. Experiments show that Agent-REINFORCE outperforms both traditional and LLM-based baselines in sample efficiency and search performance, and effectively identifies optimal graphs under joint objectives of accuracy and inference latency. Our code is available at link.

## 1 INTRODUCTION

Test-time scaling (TTS) aims to enhance large language models (LLMs) by allocating additional computational resources during inference (Brown et al., 2024; Snell et al., 2025). Prior studies have primarily investigated two architectures: (i) *parallel scaling* (Wang et al., 2023; Brown et al., 2024), which samples multiple outputs independently to increase solution diversity and aggregates them, making it suitable for tasks with uncertain or diverse solution paths; and (ii) *sequential scaling* (Madaan et al., 2023; Snell et al., 2025), which iteratively refines a single output and is well-suited for tasks that require step-by-step reasoning (see Fig. 7 (a)(b) in Appendix). Fusing the two, hybrid architectures have also been proposed, using predefined hybrid structures to combine the advantages of both (Besta et al., 2024; Snell et al., 2025) (see Fig. 7 (c) in Appendix). Despite their effectiveness, we identify two key limitations of existing TTS architectures. **First, TTS architectures are typically predefined and static, with fixed topologies across tasks.** However, our analysis shows that different tasks exhibit distinct preferences for architectural patterns, e.g., MATH favors hybrid structures, while MMLU performs better with pure parallel ones (Fig. 1(a)(c)). This suggests that architectures should adapt to task demands. **Second, existing TTS methods usually employ a single LLM for all inference steps**. In contrast, multi-LLM ensembles are preferable to leverage heterogeneous LLM skills across tasks (Jiang et al., 2023; Wang et al., 2025c). Preliminary results show that MATH benefits from mixtures of 1B–3B, whereas MMLU favors a single 8B (Fig. 1(b)(d)), underscoring the need for adaptive model selection. Overall, *test-time compute-optimal scaling* aims to maximize performance within the inference budget (Wu et al., 2025), but these findings reveal that *adaptive TTS architectures and model combinations are fundamental challenges for existing methods.*

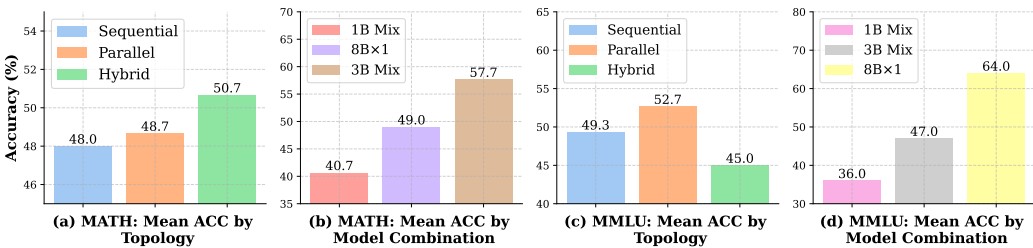

Figure 1: Accuracy across different topologies and model combinations on MATH and MMLU. LLaMA-3 models are used by default. Detailed data is in Appendix A.3.

Motivated by these observations, we study a novel problem: **searching for the compute-optimal architecture and model combination in test-time scaling for a given task**. Formally, given a task, a set of models, and a compute budget, the goal is to find the best configuration that jointly determines architecture and model assignment. Leveraging the inherent graph structure of TTS, we formulate dynamic test-time scaling as constructing a *multi-LLM collaboration graph*, where where nodes represent the chosen LLM model with assigned roles (*fuser* for parallel aggregation, *assistant* for sequential refinement), and edges denote information flow. A terminal node aggregates outputs into the final answer (see Fig. 7(d), Appendix A.1). This graph view offers a systematic foundation for dynamic optimization. However, two major challenges arise: **(i)** The search space is large due to the combinatorial choices of models and topologies, and grows rapidly with the budget. For example, with 12 nodes, the number of possible graphs ranges from $10^{18}$ to $10^{26}$ depending on model diversity (derivation in Appendix A.4). Since evaluating each candidate requires costly inference, brute-force search is infeasible. **(ii)** Tailored design requires linking task requirements to optimal TTS search patterns, which relies on an understanding of TTS behaviors. Prior work shows that performance does not grow monotonically with used budget, implying that optimal allocations are often below the maximum. These insights are key to guiding task-specific searches toward compute-optimal collaboration graphs. To address them, we conducted pilot experiments on TTS behavior analysis, which yielded three empirical insights: (1) Effective collaboration exhibits clear preferences for specific model combinations: tasks favor replication of the strongest model family, and ensembles of small models are preferred when incremental gains are substantial; (2) Both width and depth have task-dependent optima; beyond these points, extra computation will yield negative returns; (3) Graph width and depths are interdependent: growth in one dimension shifting the optimal point of the other.

We operationalize these insights by formulating the search as a probabilistic optimization problem: Learning a distribution over collaboration graphs that jointly determines edges, roles, and model assignments under a fixed budget to maximize task-specific performance. The REIN-FORCE algorithm (Williams, 1992), a gradient-based optimization method, addresses this via a *sample–gradient–update* pipeline that iteratively samples candidates, computes gradients, updates the distribution, and repeats. However, it risks local optima and its inability to incorporate empirical insights. Recent work (Liu et al., 2024a; Zhang et al., 2024a) shows that LLM-based agents are effective planners for hyperparameter optimization, with the unique advantage of leveraging external knowledge. Building on these, we propose **Agent-REINFORCE**, an LLM-agent-augmented framework for searching optimal multi-LLM collaboration graphs. Building on REINFORCE, it employs an LLM-based agent to incorporate empirical insights for candidate initialization and distribution updates, following a *sample–feedback–update* pipeline in which feedback serves as textual gradients in REINFORCE. The framework comprises three components: the `Agent`, `Archive`, and `Environment`. The `Agent` initializes promising model families and sizes guided by Insight 1 and fixes the best combination within the distribution. In subsequent stages, the new trials are sampled, the `Environment` evaluates them and returns feedback (serving as textual gradients), the `Archive` records the results, and the `Agent` updates the distribution guided by Insights 2 and 3 until convergence. By leveraging LLM-based optimization, our method efficiently identifies graphs that optimize performance alone and graphs that balance performance with inference latency under joint objectives.

Our **main contributions** are: (i) We study the novel problem of *the search for the optimal multi-LLM collaboration graph for TTS*. (ii) From three identified empirical insights in multi-LLM collaboration, we develop **Agent-REINFORCE**, an efficient LLM-guided framework for budget-constrained graph search. (iii) Experiments show that Agent-REINFORCE surpasses traditional and LLM-based baselines in search efficiency and accuracy, and effectively identifies optimal graphs under joint accuracy–latency objectives.

## 2 RELATED WORK

**Test-time Scaling and Compute-optimal Strategy.** Allocating additional compute during inference, known as *Test-Time Scaling (TTS)*, can significantly improve LLM performance (Wei et al., 2022; Wang et al., 2023; Brown et al., 2024; Wu et al., 2025). TTS methods fall into two main paradigms: *sequential scaling*, which refines outputs iteratively but risks error accumulation, and *parallel scaling*, which aggregates multiple candidates but lacks depth. Hybrid approaches (Snell et al., 2025; Wu et al., 2025) combine both but typically rely on fixed trees and a single model, limiting adaptability. *Compute-optimal TTS* seeks to allocate inference compute most effectively, revealing that small models with optimal strategies might outperform larger ones (Brown et al., 2024; Wu et al., 2025; Liu et al., 2025a; Yue et al., 2025; Snell et al., 2025; Wang et al., 2025a). Moreover, ensembles of heterogeneous models improve diversity and output quality (Jiang et al., 2023; Ashiga et al., 2025), yet remain underexplored in TTS. Motivated by this gap, we address a novel problem: unifying TTS under a graph structure that enables adaptive topologies and model combinations, and searching for compute-optimal collaboration graphs. Further discussion is provided in Appendix A.16.

**LLMs for Optimization.** LLMs, with their rich prior knowledge of machine learning and strong planning ability, have opened new opportunities for practical optimization (Zhang et al., 2025c; Guo et al., 2024). Existing research mainly falls into two categories: black-box optimization and hybrid approaches with gradient-based methods. In the black-box setting, LLMs generate and refine candidates using feedback from small training sets (Yang et al., 2024; Liu et al., 2024a; Zheng et al., 2023). Representative methods include OPRO (Yang et al., 2024), AgentHPO (Liu et al., 2024a), and GENIUS (Zheng et al., 2023), which leverage task descriptions and prior solution performance for iterative search. LLMs are particularly valuable for initialization, producing high-quality, knowledge-informed solutions that narrow the search space (Jawahar et al., 2024; Nana Teukam et al., 2025; De Zarzà et al., 2023). However, when gradient information is available, black-box approaches become inefficient due to costly evaluations. LLM-based methods address this by interleaving gradient-based training with LLM-guided exploration (Guo et al., 2024) or by generating textual guidelines as backpropagation signals (Yuksekgonul et al., 2024). Building on these advances, we extend such approaches to compute-optimal test-time scaling by optimizing a probabilistic graph with LLMs for initialization and textual parameter updates. More details are given in Appendix A.16.

## 3 PRELIMINARIES AND PROBLEM FORMULATION

**Test-time Scaling Paradigms and Their Primitives**  Test-time scaling can be broadly categorized into *parallel scaling* and *sequential scaling*. Given a query $q$ and a language model $M$ with parameters $\theta$, parallel scaling samples $k$ outputs and aggregates them via a fusion function:

$$o = f_{\text{fuse}}(\mathcal{S}, M), \quad \mathcal{S} = \{s_i \mid 1 \leq i \leq k\}, \quad s_i \sim M(s \mid q, \theta). \tag{1}$$

Sequential scaling instead performs $k$ rounds of self-refinement:

$$o = o^k, \quad o^i = f_{\text{refine}}^i(o^{i-1}, M), \quad o^0 = q. \tag{2}$$

where $f_{\text{fuse}}(\cdot)$ and $f_{\text{refine}}^i(\cdot)$ are both executed by the LLM $M$, using fusion and refinement prompts, respectively. As shown in Fig. 8, both paradigms can be decomposed into three primitives: repeated sampling, fusion, and self-refinement. Parallel scaling is repeated sampling followed by fusion; sequential scaling is iterative self-refinement. Hybrids recombines these primitives–for example, *Tree-of-Thoughts* (Yao et al., 2023) uses multi-layer repeated sampling, and *Graph-of-Thoughts* (Besta et al., 2024) integrates all three primitives in a graph.

**Multi-LLM Collaboration Graph for TTS**
Given the task-specific preference for flexible TTS paradigms beyond the predefined ones, we generalize them into a *multi-LLM collaboration graph* $G = (\mathcal{V}, \mathcal{E}, \mathbf{R}, \mathbf{M})$, where each node $v_i \in \mathcal{V}, i \in [1, n]$, represents an LLM primitive with an assigned role and model, with an example in Fig. 2. Role assignments are denoted by $\mathbf{R} = [r_1, r_2, \ldots, r_n], r_i \in \mathcal{R}$, and model assignments are denoted by $\mathbf{M} = [M_1, M_2, \ldots, M_n], M_i \in \mathcal{M}$. Thus, each node is characterized by a

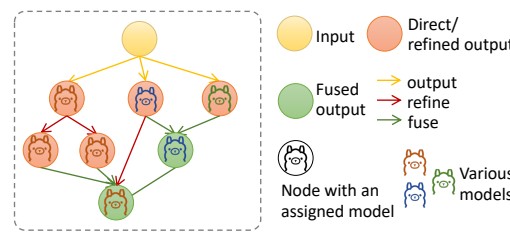

Figure 2: Generalize TTS as a graph.

role $r_i$, which specifies how it processes inputs from its predecessors, and a model $M_i$, which means which LLM is invoked. Directed edges $e_{ij} \in \mathcal{E}$ represent the flow of information from node $v_i$ to node $v_j$. We consider two roles $\mathcal{R} = \{\text{assistant}, \text{fuser}\}$, as illustrated in Fig. 2: (i) **Assistant**, which refines the outputs of its predecessors (orange nodes); and (ii) **Fuser**, which aggregates multiple predecessor outputs (green nodes). The collaboration graph $G$ is a directed acyclic graph (DAG) with a designated input node (yellow) that initiates information propagation. Message passing proceeds forward along edges until it reaches a sink node (a node without outgoing edges), whose output serves as the final prediction of the graph.

**Inference on Multi-LLM Collaboration Graph**    As illustrated in Algo. 2 in Appendix A.2, inference over a multi-LLM collaboration graph $G$ proceeds in topological order. The process begins by identifying the successor nodes of the input node. These nodes process the query to generate initial outputs that are propagated to their successors, reducing the in-degree of their successors by one accordingly. The newly activated nodes (with zero in-degree) are then executed based on their assigned roles and models. A *fuser* aggregates the outputs of its predecessors, whereas an *assistant* refines them. This procedure continues iteratively until all nodes in $G$ have been executed. The output of a unique sink node–node with no outgoing edges, is the final output of the graph.

**Budget Definition**    To enable comparative computation across models and topologies, we define the budget using a concrete compute metric, e.g., FLOPs or dollar cost. Let the computational cost of a collaboration graph $G$ be $f_{\text{cost}}(G, T)$. The budget is defined as $B = f_{\text{cost}}(G,T)/f_{\text{cost}}(G_{\text{smallest}},T)$, where $G_{\text{smallest}}$ is the single-node graph (excluding the input node) using the smallest model, corresponding to one budget unit. Thus, a multi-LLM graph with budget $B$ is equivalent to running $B$ single-node inferences on the smallest model. A detailed introduction to the budget definition is in Appendix A.8.

Formally, we report computational cost in FLOPs, which we adopt as our primary cost metric.

**Proposition 1** (FLOPs Cost Function). *For each node $v_i$, the cost depends on the size of the model and its effective input/output lengths, leading to a dependence on the node in-degree $d(v_i)$. Adding up to all nodes, the total cost can be expressed as $f_{cost}(G, T) = \sum_{v_i \in \mathcal{V}} \left[ \alpha_i \, d(v_i)^2 + \beta_i \, d(v_i) + \gamma_i \right]$, where the coefficients $\alpha_i, \beta_i, \gamma_i$ capture the contributions of the model dimension, depth, and average task input/output lengths. Detailed derivations of $\alpha_i, \beta_i, \gamma_i$ are provided in Appendix A.7.*

**Problem Definition**    The goal of *test-time compute-optimal scaling* is to allocate inference compute most effectively under a fixed budget. We formalize this as *searching for the task-specific compute-optimal multi-LLM collaboration graph*. Given training data $\mathcal{D}_{\text{train}}$, test data $\mathcal{D}_{\text{test}}$, a model pool $\mathcal{M} = \{M_1, \ldots, M_n\}$, and a budget $B$, the objective is to identify a collaboration graph that specifies role and model assignments for nodes, together with the cooperation topology, so as to maximize task performance under the budget constraint. Therefore, our research problem is defined as follows:

**Definition 1** (Test-time Compute-optimal Multi-LLM Collaboration Graph for a Specific Task). *Given the training set $\mathcal{D}_{train}$ for a given task $T$, the model pool $\mathcal{M}$, and a fixed computational budget, $B$, the goal is to identify the best collaboration graph that optimizes the performance on $\mathcal{D}_{train}$, i.e.,*

$$G^\star = \arg \max_{G \in \mathcal{G}(\mathcal{M},B)} u_T(G; \mathcal{D}_{train}) \tag{3}$$

*where $\mathcal{G}(\mathcal{M}, B) = \{G \mid f_{budget}(G, T) \leq B\}$ is the set of feasible multi-LLM collaboration graphs from $\mathcal{M}$ under budget $B$. Each $G = (\mathcal{V}, \mathcal{E}, \mathbf{R}, \mathbf{M})$ is a DAG, with node $v_i$ assigned role $r_i \in \{assistant, fuser\}$ and model $M_i \in \mathcal{M}$, and edge $e_{ij}$ denoting information flow. The utility function $u_T(G; \mathcal{D}_{train})$ measures the performance of $G$ on $\mathcal{D}_{train}$, while $G^\star$ is finally evaluated on $\mathcal{D}_{test}$.*

## 4    INSIGHTS OF MULTI-LLM COLLABORATION GRAPH FOR TTS

Searching for the optimal multi-LLM collaboration graph for test-time scaling faces two challenges: (i) the search space grows combinatorially with the increased budget, making exhaustive enumeration infeasible; and (ii) the task-specific requirements are highly specific, demanding tailored designs. We therefore conduct pilot experiments to uncover cross-task TTS patterns, which pave the way to design an efficient search method for compute-optimal collaboration graphs.

**Experimental Setting.**    We conduct preliminary experiments on three tasks: **MATH** (Hendrycks et al., 2021b) (arithmetic reasoning), **MMLU** (Hendrycks et al., 2021a) (general reasoning), and

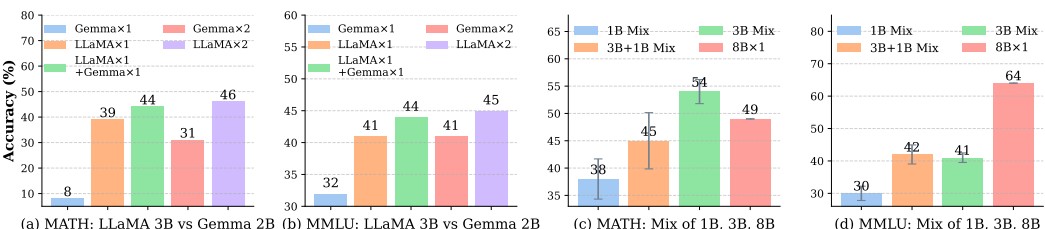

Figure 3: Performance on MATH and MMLU across model family and size. LLaMA by default.

**HumanEval** (Chen et al., 2021) (code generation), evaluated by accuracy (MATH, MMLU) and pass@1 (HumanEval). The model pool includes LLaMA-3 [1B, 3B, 8B] (Grattafiori et al., 2024) and Gemma [1B, 2B, 7B] (Team, 2025). Dataset, model, and metric details are in Appendix A.5.

**Empirical Insights on Model Selection, Parallel and Sequential Scaling.** We examine TTS behavior under increasing compute budgets and different model selections, and guide the search for the optimal multi-LLM collaboration graph in Sec.5. Fig. 3 and 4 illustrate how model selection, parallel and sequential scaling, and graph width-depth configuration influence TTS.

**Insight 1: Task-specific preferences for model family and size combinations.** We conduct preliminary tests on MATH and MMLU to examine task-specific model preferences. Results in Fig. 3(a–b) show that replicating the strongest model family is generally more effective than mixing families: for example, LLaMA consistently outperforms Gemma in the 3B space on MMLU, so using LLaMA×2 yields higher accuracy than LLaMA+Gemma or Gemma×2. Results in Fig. 3(c–d) show that within a fixed budget, reasoning tasks (MATH) benefit from ensembles of smaller models, while knowledge tasks (MMLU) prefer larger ones. These trends reflect differences in task demands and difficulty: reasoning tasks leverage multiple smaller models for iterative refinement, whereas knowledge tasks require the broader coverage of large models. A more detailed discussion is provided in Appendix A.6. Consequently, **tasks favor replication of the strongest model family, with small-model ensembles preferred only when their incremental gains are substantial.**

**Insight 2: Parallel and sequential scaling saturate and decline beyond an optimal budget.** Fig. 4(a–b) shows that both parallel (width) and sequential (depth) scaling follow a non-monotonic trend: performance improves up to a task-dependent optimum, then plateaus or declines. On MATH, for example, peak accuracy occurs at 8 parallel or 8 sequential nodes. Beyond these points, added width yields diminishing gains due to long-context limits, while added depth amplifies propagated errors. A more detailed discussion is provided in Appendix A.6. In summary, **both width and depth exhibit task-dependent optima, beyond which extra computation provides negative returns.** This insight is consistent with existing works (Wang et al., 2025b; Tang et al., 2025; Brown et al., 2024; Li et al., 2024).

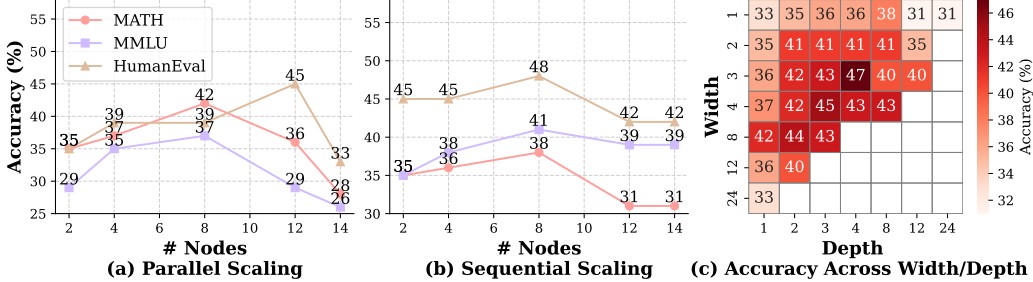

Figure 4: (a–b) Performance with Parallel and Sequential Scaling on various datasets. (c) Heatmap of performance under various Width-Depth collaboration graphs on MATH. Model is LLaMA-3 1B.

**Insight 3: Interdependence between graph width and depth.** Fig. 4(c) shows MATH performance under varying width ($w$) and depth ($d$) with $wd \leq 24$ using LLaMA-1B. Accuracy rises then falls as either dimension grows, confirming non-monotonic trends. Moreover, width and depth interact: larger widths reduce the optimal depth (e.g., 8 at $w$=1 vs. 4 at $w$=3), while deeper refinement shifts

the optimal width forward. A more detailed discussion is in Appendix A.6. Thus, **graph width and depth are interdependent, with growth in one dimension altering the optimum of the other.**

## 5 THE PROPOSED FRAMEWORK – AGENT-REINFORCE

Guided by the insights in Sec. 4, we introduce **Agent-REINFORCE**, an LLM-Agent-augmented REINFORCE algorithm that follows a *sample–feedback–update* loop to find the compute-optimal multi-LLM collaboration graph under a fixed budget. The LLM agent samples candidates and updates graphs using textual feedback (serving the textual gradient in REINFORCE) while integrating task-specific model preferences, budget allocation strategies, and width–depth interactions. We next formalize the probabilistic graph optimization problem and describe our Agent-REINFORCE.

### 5.1 PROBABILISTIC GRAPH OPTIMIZATION PROBLEM

**Optimization Problem** One way to find the optimal collaboration graph is black-box search, either through enumeration (Bergstra & Bengio, 2012) (e.g., grid or random search) or Bayesian optimization (Shahriari et al., 2015), which fits a surrogate model to the objective and selects queries via an acquisition function. Yet enumeration is infeasible as the graph space grows exponentially, while standard BO is designed for low-dimensional continuous domains and becomes sample-inefficient in large, discrete spaces. We therefore reformulate the task as a graph optimization problem, leveraging policy-gradient methods for efficient exploration, guided sampling, and budget-aware control. Given a task $T$ and its utility function $u_T$, let $G \sim \mathbb{P}_{\theta,\pi,\psi}$ denote a sampled multi-LLM collaboration graph. The distribution $\mathbb{P}_{\theta,\pi,\psi}$ is parameterized by three components: $\theta = \{\theta_{ij}\}$, where $\sigma(\theta_{ij}) \in [0,1]$ represents the probability that edge $e_{ij}$ is present; $\pi = \{\pi_i\}$, where $\mathrm{softmax}(\pi_i) \in [0,1]^{|\mathcal{R}|}$ denotes the probability of node $v_i$ selecting a role $r \in \mathcal{R}$; and $\psi = \{\psi_i\}$, where $\mathrm{softmax}(\psi_i) \in [0,1]^{|\mathcal{M}|}$ denotes the probability of node $v_i$ choosing a model $M \in \mathcal{M}$. The optimization problem is to identify

$$\theta^\star, \pi^\star, \psi^\star = \arg\max_{\theta,\pi,\psi} \mathbb{E}_{G \sim \mathbb{P}_{\theta,\pi,\psi}}\big[u_T(G, D_{\mathrm{train}})\big] \quad \text{s.t.} \quad f_{\mathrm{budget}}(G, T) \le B. \tag{4}$$

### 5.2 AGENT-REINFORCE

The REINFORCE algorithm (Williams, 1992) can optimize Eq.(4) via gradient ascent through iterative *sample–gradient–update* (sampling candidates, estimating gradients from their utility, and updating parameters; see Appendix A.11 for details). However, its step-by-step updates often lead to slow progress, local optima, and difficulty in incorporating prior insights or semantic knowledge. To overcome these limitations, we propose **Agent-REINFORCE**, an LLM-agent-augmented framework which builds on REINFORCE but replaces gradients with feedback-conditioned updates. Each iteration follows a *sample–feedback–update* loop: guided by empirical insights, the agent samples candidate graphs, receives feedback as textual gradients, and updates the distribution iteratively until convergence. As shown in Fig. 5(b–d), the framework comprises three components: Agent, Archive, and Environment. The Agent first generates candidate trials of the model family and size combinations (guided by Insight 1). Feedback from the Environment selects the

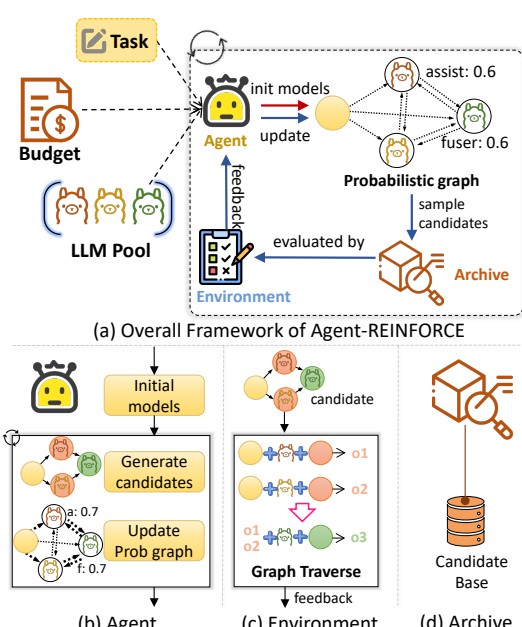

Figure 5: Overview of Agent-REINFORCE for Optimizing Collaboration Graph.

best model assignments and initializes the probabilistic graph distribution. In subsequent iterations, the Agent samples new trials from the updated distribution $\mathbb{P}_{\theta,\pi,\psi}$, the Environment evaluates them, and the Archive records results. The Agent then updates the distribution based on feedback and history, and this loop continues until convergence. The full procedure is given in Algo. 1.

---

**Algorithm 1** Agent-REINFORCE: Compute-Optimal Collaboration Graph Optimization

---

**Require:** Task $T$, model set $\mathcal{M}$, agent $\mathcal{A}$, environment $\mathcal{E}$, budget $B$
**Ensure:** Optimized collaboration graph $G$
 1: Initialize archive $\mathcal{L} \leftarrow \emptyset$
 2: Stage 1: $\mathcal{C} \leftarrow \mathcal{A}.\texttt{select\_family\_size}(T, \mathcal{M}, B); \mathcal{S} \leftarrow \mathcal{E}.\texttt{execute}(\mathcal{C})$       (Init. Stage 1)
 3: Stage 2: $\mathcal{C} \leftarrow \mathcal{A}.\texttt{select\_instance}(T, \mathcal{M}, \mathcal{S}, B); \mathcal{S} \leftarrow \mathcal{E}.\texttt{execute}(\mathcal{C})$       (Init. Stage 2)
 4: Initialize nodes in $\tilde{\mathbf{G}}$ with the best model family, size, and instance count       (Insight 1)
 5: **while** stopping criterion not met **do**       (Subsequent stages)
 6:     Update archive $\mathcal{L} \leftarrow \mathcal{L} \cup \{(\mathcal{C}, \mathcal{S}, \tilde{\mathbf{G}})\}$
 7:     Sample new trials $\mathcal{C} \leftarrow \mathcal{A}.\texttt{sampling}(\tilde{\mathbf{G}}, B)$
 8:     Get feedback (textual gradient) $\mathcal{S} \leftarrow \mathcal{E}.\texttt{execute}(\mathcal{C})$
 9:     Update graph $\tilde{\mathbf{G}} \leftarrow \mathcal{A}.\texttt{update}(\mathcal{C}, \mathcal{S}, \mathcal{L}, \tilde{\mathbf{G}})$       (Insights 2,3)
10: **end while**
11: **return** Graph $G$ by deterministic decoding from $\tilde{\mathbf{G}}$

---

**Agent component.** The LLM-base `Agent`, in Fig. 5 (b), initializes model assignments, samples new trials, and updates the probabilistic graph. Since LLMs lack prior knowledge of test-time scaling, which is relatively new, we incorporate Insight 1 to guide the initialization of model assignments, and Insights 2 and 3 to inform subsequent updates. *Insight 1* shows that tasks prefer replicating the strongest family, with small-model ensembles chosen when their gains are high. Hence, initialization focuses on task-specific model assignments (family, size, and instances) to guide optimization and reduce wasted exploration. We initialize the family-size and instance counts in two stages.

First, the `Agent` identifies family and size preferences using each model's meta-information from HuggingFace (hug), including prior performance and the task description. Prior performance guides family selection; when unavailable, initial trials pre-test each model's prior performance to infer family preferences. For size selection, the incremental gains from ensembling one versus two small models relative to a single large model inform size preference, motivating trials that explore both small ensembles and large models. Therefore, the agent initializes candidates as $\mathcal{A}.\texttt{select\_family\_size}(T, \mathcal{M}, B)$, retaining only those within budget $B$, and obtains performance scores from the `Environment` as feedback $\mathcal{S}$ to identify the preferred family and size (Algo. 1, Line 2).

Second, using feedback $\mathcal{S}$, the `Agent` generates diverse candidate model combinations within budget $B$ via $\mathcal{A}.\texttt{select\_instance}(T, \mathcal{M}, \mathcal{S}, B)$, prioritizing the selected family and size while varying instance counts. For each candidate, graph topologies and role assignments are randomly sampled (Algo. 1, Line 3). Feedback is averaged, and the best configuration, covering family, size, and instances, initializes the graph (Algo. 1, Line 4). In subsequent stages, nodes retain the model assignments, while edges and roles are sampled from the probabilistic graph $\tilde{\mathbf{G}}$ via $\mathcal{A}.\texttt{sampling}(\tilde{\mathbf{G}}, B)$.

*Insight 2* shows that width and depth have task-specific optima: performance improves with more nodes up to a point, then degrades. We incorporate this into the update prompt ($\mathcal{A}.\texttt{update}$, Algo. Line 9), guiding the `Agent` to "identify the optimization direction for finding the optimal width and depth" by leveraging feedback from current and past trials to adjust the probabilistic graph toward the optimal width–depth balance and accelerate convergence.

*Insight 3* highlights the interdependence between width and depth: under a fixed budget, improving one often requires reducing the other. To manage this, we embed an instruction into the update prompt ($\mathcal{A}.\texttt{update}$, Algo. Line 9) that directs the `Agent` to exploit the LLM's planning ability to explore these trade-offs between width and depth and adaptively identify critical graphs within budget.

The instructions derived from the insights are applied continuously during the optimization process. Based on the feedback, the `Agent` updates the probabilistic graph (Algo. Line 9), which is then used to sample the next batch of trials (Algo. Line 7). The prompt design is provided in Appendix A.12.

**Environment & Archive Components.** `Environment` converts candidate graphs from the `Agent` into executable scripts, runs them in the actual task platform on a small training batch, and returns performance feedback (Fig. 5c; Algo. 1, Lines 2–3,8). `Archive` stores the probabilistic graph, sampled trials, and corresponding feedback (Fig. 5d; Algo. 1, Lines 1,6), tracking the optimization process across iterations and providing historical traces for the `Agent` to refine future updates.

Table 1: Performance across MATH, MMLU, and HumanEval at 80 budget. Acc (↑) means Accuracy (%, higher is better), Sear. (↓) means total search time in seconds (lower is better), and Inf. (↓) means average inference time in seconds per test query (lower is better). Best in each column is bolded.

| Method | MATH | | | MMLU | | | HumanEval | | | Average | | |
|---|---|---|---|---|---|---|---|---|---|---|---|---|
| | Acc | Sear. | Inf. | Acc | Sear. | Inf. | P@1 | Sear. | Inf. | Sco. | Sear. | Inf. |
| Single Model | 49 | - | - | 64 | - | - | 60 | - | - | 58 | - | - |
| Random | 39 | 2852 | 28.5 | 44 | 658 | 6.6 | 63 | 1560 | 47.3 | 49 | 1690 | 27.5 |
| BO | 42 | 3076 | 30.8 | $36.8 \pm 5.2$ | 2150 | 21.5 | 33 | 2588 | 78.4 | 38 | 2605 | 43.6 |
| GPTSwarm | 40 | 943 | 9.4 | 42 | **463** | **4.6** | 55 | 804 | 24.4 | 46 | 737 | 12.8 |
| MaaO | 34 | 1440 | 14.4 | 41 | 738 | 7.4 | 42 | 860 | 26.1 | 39 | 1013 | 16.0 |
| TextGrad | 41 | 3687 | 36.9 | $39.9 \pm 3.6$ | 2276 | 22.8 | 42 | 2842 | 86.1 | 43 | 2935 | 48.6 |
| Ours | **56** | **804** | **8.0** | $\mathbf{61.5 \pm 5.1}$ | 493 | 4.9 | **73** | **300** | **9.1** | **61** | **532** | **7.3** |

# 6 EXPERIMENTS

This section evaluates Agent-REINFORCE for compute-optimal collaboration graphs in TTS, covering ablations, varying budgets, joint objectives, alternative budget metrics, and visualizations.

**Experimental Setup.** We experiment on MATH, MMLU, and HumanEval using LLaMA models (1B-8B) (Grattafiori et al., 2024) and Gemma models (1B-7B) (Team, 2025) (details in Appendix A.5). Baselines fall into three groups: (i) traditional: Bayesian Optimization (BO) (Jones et al., 1998; Shahriari et al., 2015) and random search; (ii) gradient-based: GPTSwarm (Zhuge et al., 2024), a REINFORCE framework with gradient updates, and MaaO (Guo et al., 2024), combining gradient training with LLM guidance; and (iii) LLM-based: TextGrad (Yuksekgonul et al., 2024), which relies solely on textual guidelines. As these methods are not tailored to our setting, we adapt them for test-time compute-optimal graph search (details in Appendix A.13). All methods are run for up to 30 search iterations on the training data and use the validation set to determine convergence. Search is stopped if the average validation performance does not improve for 10 iterations. The final searched graph is evaluated on the test set. We use DeepSeek-R1 (Guo et al., 2025) as the LLM search agent.

**Main Results.** Tab. 1 reports test performance and convergence time, and Fig. 9 in the Appendix shows training trajectories. We observe: (1) Our method achieves the highest average test-set score (higher accuracy or Pass@1) while converging substantially faster (lower search time). This is enabled by Insights 2–3, which guide the search toward promising regions, and Insight 1, which provides a strong initialization and avoids wasted trials. (2) Compared with the LLM-based TextGrad, our method is much more efficient by pruning high-latency candidates early. Among the methods, TextGrad yields the highest inference latency in the searched graphs, reflecting its tendency to favor dense connections or larger node counts that drive full-budget utilization. Such usage often produces high-overhead graphs and consequently slower convergence. (3) The gradient-based GPTSwarm and MaaO converge quickly but often produce graphs inferior even to random search, due to their vulnerability to local optima. This underscores the importance of combining global exploration with local refinement. (4) The traditional Bayesian optimization method also suffers from local optima and slow updates due to a lack of task-specific guidance. Random search shows some robustness and can occasionally find competitive solutions, but it remains inefficient and unstable.

**Ablation Studies.** We evaluate the contribution of each insight through ablation, comparing the full method with variants: w/o Insight 1 uses random initialization instead of task- and model-informed initialization, while w/o Insight 2/3 removes prompt components for budget optima and width–depth dependencies. Tab. 2 shows that removing any insight slows convergence by generating inefficient graphs; w/o Insight 1 enlarges the candidate space, and w/o Insight

Table 2: Ablation study of Agent-REINFORCE on MATH and MMLU w/o insights and role setting.

| Methods | MATH | | MMLU | |
|---|---|---|---|---|
| | Acc | Sear. | Acc | Sear. |
| Agent-REINFORCE | 56 | 804 | 54 | 493 |
| w/o Insight 1 | 45 | 1946 | 42 | 1293 |
| w/o Insight 2 | 49 | 2208 | 47 | 896 |
| w/o Insight 3 | 48 | 1436 | 54 | 487 |
| w/o Role | 52 | 785 | 54 | 677 |

2/3 biases exploration toward high-budget graphs. Performance drops most under w/o Insight 1, as random initialization yields suboptimal starts that limit later search. Excluding Insight 2 or 3 also reduces accuracy by losing guidance on budget and width–depth trade-offs. We also perform an ablation by removing role setting, letting all nodes process predecessors' outputs and generate new answers, which degrades graph performance on MATH. This highlights the importance of the *fuser–assistant* role division in test-time scaling. We note that MMLU performance remains stable without Insight 3 or role settings, as it favors larger models with fewer nodes, reducing the impact of width–depth trade-offs and roles.

Table 3: MATH Acc, Sear, and Inf under various FLOPs and price budget.

| Method | Price ≤ $5E-4 | | | FLOPs Budget 42 | | | FLOPs Budget 18 | | |
|---|---|---|---|---|---|---|---|---|---|
| | Acc | Sear. | Inf. | Acc | Sear. | Inf. | Acc | Sear. | Inf. |
| Random | 35 | 2546 | 52.5 | 33 | 1706 | 56.9 | 39 | 1440 | 16.0 |
| BO | 36 | 2372 | 56.6 | 45 | 2724 | 49.5 | 38 | 1634 | 23.1 |
| GPTSwarm | 43 | 832 | 20.5 | 44 | 858 | 31.2 | 44 | 1028 | 29.1 |
| MaaO | 47 | 1104 | 20.0 | 46 | 889 | 50.9 | 44 | 836 | **14.6** |
| TextGrad | 22 | 3062 | 57.9 | 45 | 2661 | 48.6 | 40 | 2553 | 16.8 |
| Ours | **56** | **648** | **18.1** | **50** | **726** | **11.5** | **47** | **771** | 16.7 |

**Performance Under Various Budget Settings.** We evaluate search performance on the MATH dataset under FLOPs budgets of 18 and 42, accommodating $1 \times 8B$ and $[2,3] \times 8B$ models, respectively. As shown in Tab. 3, our method consistently delivers superior efficiency and accuracy, demonstrating strong generalization. Notably, some baselines perform better at smaller budgets (e.g., MaaO: 44 at budget 18 vs. 34 at 80) because they overlook that the optimal budget is often below the maximum. As noted in Insight 2, computation beyond the optimum yields negative returns, whereas smaller budgets closer to the budget optimum can bring these methods nearer to peak performance.

**Latency-aware Joint Optimization Objective.** To demonstrate our method's ability to handle joint optimization objectives, we optimize both performance and latency through multidimensional feedback, achieving a balance between accuracy and efficiency. The details of the optimization with a joint objective are in Appendix A.9. On MATH with a 42 FLOPs budget, the searched graph achieves an average latency of 3.1 seconds per test query, which is much lower than the 11.5 seconds under a performance-only objective, thereby validating its effectiveness for multi-objective optimization, even though performance decreases slightly from 50 to 46.

**Generalization to the Dollar Cost as Budget.** Beyond FLOPs, end-users often care about the monetary cost of API calls. We introduce price as an additional budget metric, directly measured in currency units. As shown in Tab. 5 (Appendix A.10) , cost scales with input and output tokens, so $f_{\text{budget}}(G, T)$ is redefined as input length times per-token input price plus output length times per-token output price. Under a fixed API budget $5E-4 per query (from $4 \times 8B$ to $6 \times 8B$ models), the results in Tab. 3 show Agent-REINFORCE excels in both accuracy and efficiency, showing strong generalization across cost metrics.

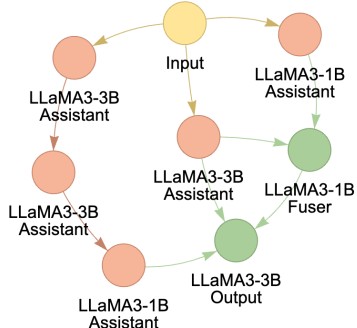

Figure 6: Optimal graph on MATH.

**Visualization** Fig. 6 visualizes the optimal collaboration graph within the budget 80 for the MATH task. The result indicates a clear preference for small-model ensembles, as the relatively low task difficulty enables small models to meet performance requirements, while additional instances further enhance their effectiveness. The structure favors a hybrid scaling biased toward sequential refinement (width 3, depth 4), since multi-step math reasoning benefits from iterative self-refinement, which sequential structures are better suited to support.

# 7 CONCLUSION

We study a novel problem of searching task-specific, compute-optimal test-time scaling over multi-LLM collaboration graphs under a fixed budget, with an exponentially large design space in model choices and nodes. From pilot analysis, we gain three empirical insights: (1) tasks replicate the strongest model family, with small-model ensembles favored when incremental gains are high; (2) width and depth admit task-specific optima, beyond which additional compute degrades performance; and (3) width and depth interact, with growth in one shifting the optimum of the other. Based on these findings, we propose **Agent-REINFORCE**, an LLM-agent framework that conducts budget-aware, feedback-driven search on collaboration graphs. Experiments show that our proposed method outperforms traditional and LLM-based baselines in search efficiency and performance, while also showing the ability to find optimal graphs under a joint performance-latency objective.

## ETHICS STATEMENT

This work adheres to the ICLR Code of Ethics. No human or animal subjects were involved. Datasets (MATH, MMLU, HumanEval) were used in compliance with guidelines, with no privacy violations or personally identifiable information. We ensured fairness, avoided bias, and upheld transparency and integrity throughout the research.

## REPRODUCIBILITY STATEMENT

We provide an anonymous code package and configures at link to ensure reproducibility of all experiments; Training/inference details are provided in Section 6; The datasets we used are public datasets, with sources, task setups and pre-processing steps provided in Appendix A.5; The prompt design in AGENT-REINFORCE is detailed in Appendix A.12, and the method internals are given in Appendix A.2 and Appendix A.11. Any additional insights and related works are summarized in Appendix A.6 and Appendix A.16.

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

## A  APPENDIX

### A.1  TEST-TIME SCALING: MODES AND BUILDING BLOCKS

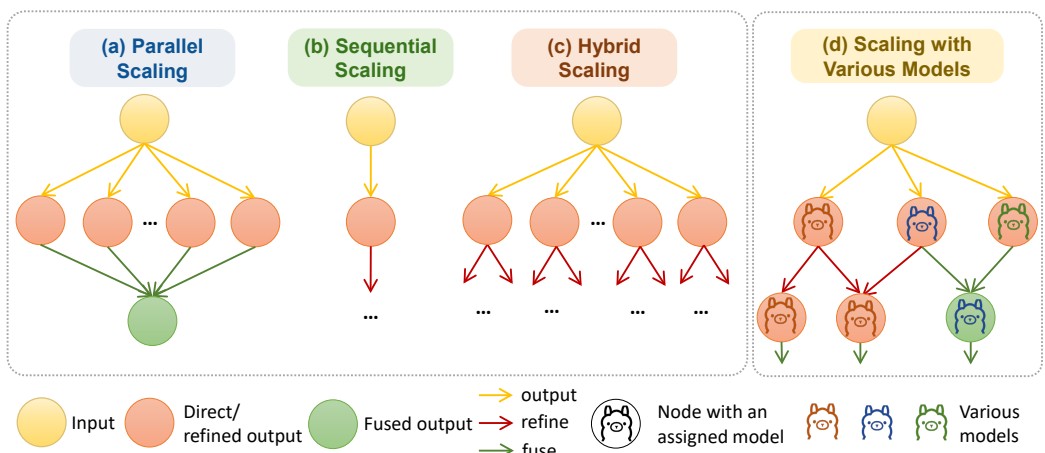

Figure 7: Test-Time Scaling Paradigms: (a–c) Fixed topologies with single-model assignments, and (d) dynamic scaling with diverse models.

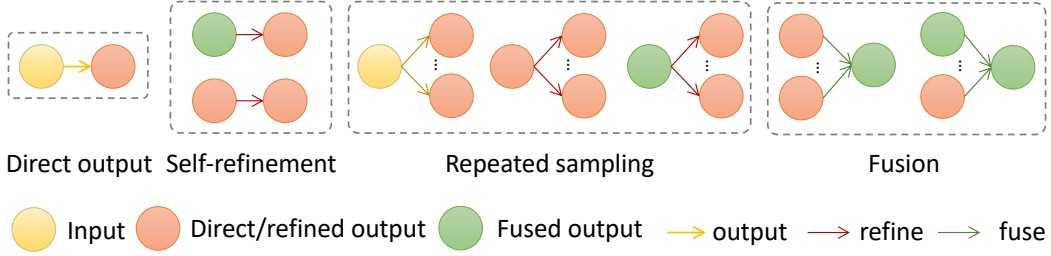

Figure 8: Test-time scaling primitives.

Fig. 7 shows four paradigms: (a) parallel via repeated sampling + aggregation; (b) sequential via iterative self-refinement; (c) fixed hybrids that fuse both; and (d) our dynamic setting that searches architectures and assigns heterogeneous models under a compute budget. Fig. 8 reduces these to three primitives—repeated sampling, fusion, self-refinement, and frames dynamic TTS as a multi-LLM collaboration graph with role-assigned nodes (e.g., fuser, assistant), directed information flow, and a terminal aggregator.

### A.2  INFERENCE ON MULTI-LLM COLLABORATION GRAPH FOR TTS ALGORITHM

Algo 2 executes the collaboration graph $G$ in topological order: Successors of the input node generate initial outputs; nodes activate when in-degree reaches zero and run by role—*fuser* (aggregate) or *assistant* (refine)—propagating results forward. The unique sink node produces the final answer.

### A.3  PILOT EXPERIMENTS FOR EXISTING TTS

Table 4 summarizes the task-specific preferences for topologies and model combinations. In MATH, the hybrid graph topology combined with a mixture of 3B models yields the best accuracy. In contrast, MMLU shows a clear preference for pure parallel graph topologies and the use of a single 8B model. These results indicate that different tasks exhibit distinct preferences for architectural patterns and model configurations.

---

**Algorithm 2** Inference on Multi-LLM Collaboration Graph for TTS

---

**Require:** Query $q$, graph $G = (\mathcal{V}, \mathcal{E}, \mathbf{R}, \mathbf{M})$ (DAG with a unique sink $v_{\text{sink}}$)
**Ensure:** Final output $o$
  1: Initialize $d_{\text{in}}(v), d_{\text{out}}(v)$ and buffers $\mathcal{O}(v) \leftarrow \emptyset$ for all $v \in \mathcal{V}$
  2: $\mathcal{Q} \leftarrow \{ v \in \mathcal{V} \mid d_{\text{in}}(v) = 0 \}$                                      ▷ topological frontier
  3: **while** $\mathcal{Q} \neq \emptyset$ **do**
  4:     Remove a node $v$ from $\mathcal{Q}$
  5:     $\mathcal{C} \leftarrow \bigcup_{u \in \text{pred}(v)} \mathcal{O}(u)$
  6:     **if** $r_v = $ fuser **then**
  7:         $\mathcal{O}(v) \leftarrow f_{\text{fuse}}(q, \mathcal{C}, M_v)$
  8:     **else**                                            ▷ $r_v = $ assistant
  9:         $\mathcal{O}(v) \leftarrow f_{\text{refine}}(q, \mathcal{C}, M_v)$
10:     **end if**
11:     **for all** $w \in \text{succ}(v)$ **do**
12:         $d_{\text{in}}(w) \leftarrow d_{\text{in}}(w) - 1$; **if** $d_{\text{in}}(w) = 0$ **then** add $w$ to $\mathcal{Q}$
13:     **end for**
14: **end while**
15: **return** $o \leftarrow \mathcal{O}(v_{\text{sink}})$                        ▷ unique sink with $d_{\text{out}}(v_{\text{sink}}) = 0$

---

Table 4: Accuracy (ACC, %) across different topologies and model combinations on MATH and MMLU. LLaMA-3 models are used by default. Results are averaged over 10 random graphs.

| Dataset | Model Comb. | Sequential | Parallel | Hybrid |
|---------|-------------|------------|----------|--------|
| MATH | 1B Mix | 37 | 41 | 44 |
| | 3B Mix | 58 | 56 | 59 |
| | 8B×1 | 49 | 49 | 49 |
| MMLU | 1B Mix | 35 | 43 | 30 |
| | 3B Mix | 49 | 51 | 41 |
| | 8B×1 | 64 | 64 | 64 |

## A.4 CALCULATION OF THE NUMBER OF DAGs

Given $n$ nodes, the spectrum of possible configurations ranges from *totally indistinguishable* nodes to *totally distinguishable* nodes. The number of directed acyclic graphs (DAGs) lies within this range: the indistinguishable case corresponds to counting the number of *non-isomorphic DAGs* (where isomorphic topologies are counted only once), while the distinguishable case corresponds to counting the number of *labeled DAGs*.

**Indistinguishable nodes: Number of non-isomorphic DAGs.** A closed-form characterization can be derived from the fact that every DAG admits at least one topological ordering. If we fix the order $1 < 2 < \cdots < n$, then only edges of the form $i \to j$ with $i < j$ are permitted. This yields $\frac{n(n-1)}{2}$ possible edges, and thus $2^{\binom{n}{2}}$ candidate adjacency matrices, all acyclic by construction. However, many of these candidates are *isomorphic*. To correctly count non-isomorphic DAGs, each candidate graph is reduced to a *canonical labeling*, and graphs with the same canonical form are merged. To reduce the cost of considering all permutations, nodes are grouped by their in-degree and out-degree, and permutations are applied only within these groups, which substantially reduces computational complexity. The Python implementation in Listing 1 computes the number of non-isomorphic DAGs.

**Distinguishable nodes: Number of labeled DAGs.** When nodes are labeled, the total number of DAGs can be computed using a well-known recurrence relation: $A(0) = 1$, $A(n) = \sum_{k=1}^{n} (-1)^{k+1} \binom{n}{k} 2^{k(n-k)} A(n-k)$. Here, $A(n)$ denotes the number of labeled DAGs on $n$ nodes. This formulation accounts for all possible edge configurations under node labeling and ensures that only acyclic structures are counted. The corresponding Python implementation is provided in the Listing 2.

```python
import itertools as it

def upper_adj_bitmasks(n, bits):
    rows = [0]*n
    for i in range(n):
        for j in range(i+1, n):
            if bits & 1: rows[i] |= (1<<j)
            bits >>= 1
    return rows

def indegree_outdegree(rows):
    n = len(rows)
    outdeg = [r.bit_count() for r in rows]
    indeg = [0]*n
    for i,r in enumerate(rows):
        while r:
            j = (r&-r).bit_length()-1
            indeg[j] += 1
            r &= r-1
    return tuple(zip(outdeg, indeg))

def permute_rows(rows, perm):
    inv = [0]*len(perm)
    for i,p in enumerate(perm): inv[p]=i
    return [sum(1<<inv[j] for j in range(len(rows)) if (rows[perm[i]]>>j)&1) for i in range(
        len(rows))]

def canonical_form_rows(rows):
    degs = indegree_outdegree(rows)
    groups = {}
    for i,deg in enumerate(degs): groups.setdefault(deg,[]).append(i)
    perms = [it.permutations(g) for g in groups.values()]
    best = None
    for p in it.product(*perms):
        perm = [x for part in p for x in part]
        newrows = permute_rows(rows, perm)
        key = ''.join('1' if (newrows[i]>>j)&1 else '0' for i in range(len(rows)) for j in
            range(len(rows)))
        if best is None or key < best: best = key
    return best

def count_unlabeled_dags(n):
    m = n*(n-1)//2
    return len({canonical_form_rows(upper_adj_bitmasks(n,b)) for b in range(1<<m)})

for n in range(1,8): print(n, count_unlabeled_dags(n))

# Results (number of non-isomorphic DAGs)
# n=1: 1
# n=2: 2
# n=3: 8
# n=4: 54
# n=5: 762
# n=6: 21,542
# n=7: 1,259,209
```

Listing 1: Python code computes the number of non-isomorphic DAGs

```python
import math
from functools import lru_cache

@lru_cache(None)
def labeled_dags(n):
    if n==0:
        return 1
    s=0
    for k in range(1,n+1):
        s += (-1)**(k+1) * math.comb(n,k) * (2**(k*(n-k))) * labeled_dags(n-k)
    return s

for n in range(1,9):
    print(n, labeled_dags(n))

# n=8: 783,702,329,343
```

Listing 2: Python code computes the number of labeled DAGs

## A.5 Tasks, Datasets, and Models

**MATH dataset (Hendrycks et al., 2021b)** The MATH dataset is used for *arithmetic reasoning* evaluation tasks, consisting of 12,500 competition-level problems from high school contests. Each problem is accompanied by a step-by-step solution, which supports evaluation of final-answer *accuracy* as the primary metric. Serving as a rigorous benchmark for symbolic manipulation and multi-step mathematical reasoning, MATH is widely used to test the limits of language models. In our experiments, we sample 750 problems for training and 100 for testing, with average prompt and generation lengths of 202 and 275 tokens, respectively.

**MMLU dataset (Hendrycks et al., 2021a)** The Massive Multitask Language Understanding (MMLU) dataset is a comprehensive benchmark for evaluating *knowledge and general reasoning tasks* across 57 tasks spanning humanities, social sciences, STEM, and professional fields, with questions ranging from elementary to advanced difficulty. Each task is presented in a multiple choice format and *precision* is used as a standard evaluation metric. MMLU has become a widely adopted benchmark for assessing the general knowledge and cross-domain adaptability of large language models. In our experiments, we randomly sampled 285 questions for training and 100 for testing, with average prompt and generation lengths of 213 and 230 tokens, respectively.

**HumanEval dataset (Chen et al., 2021)** The HumanEval dataset is a benchmark designed to assess *code generation and synthesis* capabilities of language models. It contains 164 Python programming problems, each consisting of a function signature, natural language docstring, and unit tests for automatic evaluation. The primary metric is *passk*, which measures the probability that at least one of $k$ generated solutions passes all hidden test cases. HumanEval has become a standard benchmark for evaluating the ability of models to translate natural language descriptions into correct, executable code. In our experiments, we randomly sample 128 instances for training and others for testing, with average prompt and generation lengths of 181 and 104 tokens, respectively.

**Language models adopted** We evaluate our method using language models of varying scales from the LLaMA-3 family (Grattafiori et al., 2024) and Gemma family Team (2025). To promote diversity in generations and enhance coverage during parallel sampling, we set the decoding temperature to 0.9 while retaining all other hyperparameters at their default values. All experiments are conducted on an NVIDIA A800 GPU with 80GB HBM3 memory to ensure a consistent runtime environment.

## A.6 Detailed Insights

**Insight 1: Task-specific preferences for model family and size combinations.** We conduct preliminary tests across various combinations of model families and sizes on the MATH and MMLU datasets to explore the task-specific model preferences. Fig. 3(a–b) compares performance with various family combinations. The results show that allocating the budget to multiple instances of the strongest model is more effective than mixing families. For example, within the 3B space of LLaMA and Gemma on MMLU, LLaMA outperforms Gemma; thus, LLaMA×2 surpasses both Gemma+LLaMA and Gemma×2. This is because test-time scaling effectiveness is driven by the capability of base models, favoring replication of stronger ones. Fig. 3 (c–d) reports 10-run average performance with 90% confidence intervals under the same limited FLOPs budget, considering LLaMA 1B, 3B, and 8B, to explore whether limited budgets should be allocated to more small models or fewer large models (noting that with an unlimited budget, large models are always optimal). Reasoning tasks (MATH) favor mixtures of smaller models (3B×3), while knowledge tasks (MMLU) prefer larger models (8B×1). The trade-off depends on marginal performance gains: on MATH, LLaMA 3B improves by 7 points (from 39% to 46%) when scaled from one to two instances, showing the potential to surpass a single 8B (49%) with more instances, thus favoring small-model mixtures; on MMLU, the gain (41% to 45%) is modest, making larger models (8B×1 with 64%) preferred. These are attributed to (i) task demands: reasoning tasks benefit from smaller-models ensembles because multiple models provide more opportunities to refine the answers with multi-step reasoning, whereas knowledge tasks need broad parametric knowledge coverage, better supported by large models; and (ii) task difficulty: easier tasks yield larger gains from small models, as they can already solve such tasks well and scaling further improves performance, whereas harder tasks are challenging for small models and demand large ones. Consequently, **tasks favor replication of the strongest model family, with small-model ensembles preferred only when their incremental gains are substantial.**

**Insight 2: Parallel and sequential scaling saturate and decline beyond an optimal budget.** Fig. 4 (a–b) shows that both parallel and sequential scaling on various datasets follow a non-monotonic pattern. Increasing the number of parallel nodes (width) or sequential nodes (depth) initially improves performance, but beyond a task-specific optimal point, performance plateaus and eventually declines. For example, peak performance is achieved at 8 parallel nodes or 8 sequential nodes on MATH, after which additional nodes yield no consistent gains. This performance degradation arises from different sources. In parallel scaling, performance converges once a sufficient width ensures dominance of correct answers, so additional nodes provide little benefit. Excessive outputs from preceding nodes lengthen input contexts, straining long-context capacity and degrading performance. In sequential scaling, performance improves while refinement benefits exceed potential propagated errors; once the refinement capacity is reached, additional steps mainly propagate and amplify errors, leading to performance degradation. In summary, **both width and depth exhibit task-dependent optima, beyond which extra computation provides negative returns.**

**Insight 3: Interdependence between graph width and depth.** Fig. 4 (c) shows MATH performance under varying width (parallel nodes) and depth (sequential nodes) combinations. We adopt a fixed architecture that first performs parallel sampling of $w$ nodes from the input node, followed by sequential self-refinement of $d$ nodes for each sampled branch, using the LLaMA-1B model uniformly across all nodes. To examine the trade-off between width and depth, we impose the constraint $wd \leq 24$. We observe: (i) accuracy at the optimal depth rises then falls as width increases (e.g., 38 at width 1, 47 at width 3, 45 at width 4), consistent with Insight 2; (ii) the optimal depth decreases with larger widths (e.g., 8 at width 1 vs. 4 at width 3), as initially wider structures enhance refinement capacity and accelerate convergence. Increasing depth yields the same pattern on width: accuracy follows a non-monotonic trend, and the optimal width decreases because deeper refinement allows correct answers to dominate earlier, shifting the optimal width point forward. In summary, **graph width and depth are interdependent, with growth in one dimension shifting the optimal point of the other.**

## A.7 $f_{\text{COST}}(G, T)$ WITH THE FLOPs COMPUTE METRIC

We adopt a simplified but standard FLOPs accounting scheme, where one multiply-add counts as 2 FLOPs, and causal self-attention reuses cached keys/values during decoding. Consider a model at node $v$ with non-embedding model parameters $M$, hidden size $D$, and layers $L$. Let $N_p$ and $N_d$ denote the input (prefill) and output (decode) lengths for node $v$ on task $T = (\bar{N}_p^T, \bar{N}_d^T)$ where $\bar{N}_p^T$ and $\bar{N}_d^T$ are the average length of input and output, respectively.

**Token-wise projection/MLP FLOPs.** Each non-embedding weight is applied once per token through a matrix multiplication followed by addition, yielding approximately $2M$ FLOPs per token. Aggregating across sequence lengths, we obtain $2MN_p$ for prefill, $2MN_d$ for decode.

**Attention FLOPs.** For a single layer and a single head, the number of attention score dot-products (queries × keys) is:

- **Prefill (length $N_p$):** causal masking yields a triangular count

$$\sum_{i=1}^{N_p} i = \frac{N_p(N_p+1)}{2}.$$

- **Decode (length $N_d$):** token $t$ attends to $N_p + t$ tokens, giving

$$\sum_{t=1}^{N_d} (N_p + t) = N_d N_p + \frac{N_d(N_d+1)}{2} = \frac{N_d(2N_p + N_d + 1)}{2}.$$

Since each attention requires both query-key dot products and value applications, the total multiply-adds are $4LD$ FLOPs per token. Summing across $D$ hidden size and $L$ layers gives

$$\text{FLOPs}_{\text{attn, prefill}} = 2LD\,N_p(N_p + 1), \qquad \text{FLOPs}_{\text{attn, decode}} = 2LD\,N_d(2N_p + N_d + 1).$$

These formulas combine constant factors from scoring, softmax, and value multiplication, while preserving quadratic and linear scaling in $N_p$ and $N_d$.

**Node-level cost.** Summing the projection/MLP and attention costs yields

$$f_{\text{cost\_prefill}}(N_p, M) = 2MN_p + 2LD\, N_p(N_p + 1),$$

$$f_{\text{cost\_decode}}(N_p, N_d, M) = 2MN_d + 2LD\, N_d(2N_p + N_d + 1),$$

so that

$$f_{\text{cost}}(N_p, N_d, M) = f_{\text{cost\_prefill}}(N_p, M) + f_{\text{cost\_decode}}(N_p, N_d, M).$$

**Effective input length in a collaboration graph.** In a multi-LLM collaboration graph $G = (\mathcal{V}, \mathcal{E}, \mathbf{R}, \mathbf{M})$, the effective prefill length for node $v_i$ depends on the task average input and the number of predecessor outputs concatenated to its input. With $T = (\bar{N}_p^T, \bar{N}_d^T)$ and in-degree $d(v_i)$, we set

$$N_p^{v_i} = \bar{N}_p^T + d(v_i)\, \bar{N}_d^T, \qquad N_d^{v_i} = \bar{N}_d^T.$$

**Graph-level cost.** Summing node costs across the graph,

$$f_{\text{cost}}(G, T) = \sum_{v_i \in \mathcal{V}} \Big[ f_{\text{cost\_prefill}}\big(N_p^{v_i}, M_i\big) + f_{\text{cost\_decode}}\big(N_p^{v_i}, N_d^{v_i}, M_i\big) \Big].$$

Substituting node-level cost formulas,

$$f_{\text{cost}}(G, T) = \sum_{v_i \in \mathcal{V}} \Big[ 2M_i N_p^{v_i} + 2L_i D_i\, N_p^{v_i}(N_p^{v_i} + 1) + 2M_i N_d^{v_i} + 2L_i D_i\, N_d^{v_i}\big(2N_p^{v_i} + N_d^{v_i} + 1\big) \Big].$$

**Simplified form.** Let $A = \bar{N}_p^T$, $B = \bar{N}_d^T$, and $d_i = d(v_i)$. Then

$$f_{\text{cost}}(G, T) = \sum_{v_i \in \mathcal{V}} \Big[ 2M_i(A + d_i B) + 2L_i D_i(A + d_i B)(A + d_i B + 1) + 2M_i B + 2L_i D_i B\big(2(A + d_i B) + B + 1\big) \Big].$$

Expanding and grouping by $d_i$ yields a quadratic form

$$f_{\text{cost}}(G, T) = \sum_{v_i \in \mathcal{V}} \big[ \alpha_i\, d_i^2 + \beta_i\, d_i + \gamma_i \big],$$

with coefficients

$$\alpha_i = 2L_i D_i B^2, \quad \beta_i = 2M_i B + 2L_i D_i B(2A + 2B + 1), \quad \gamma_i = 2(M_i + L_i D_i)(A + B) + 2L_i D_i(A + B)^2.$$

Please remark that

(i) **Verifier/top-$k$ filtering.** If a fuser applies top-$k$ selection on predecessor outputs, replace $d(v_i)$ by $\min\{d(v_i), k\}$ in $N_p^{v_i}$.

(ii) **Alternative metrics.** For monetary cost, replace FLOPs-based node terms with calibrated surrogates $\{f_{\text{cost\_prefill}}, f_{\text{cost\_decode}}\}$ per model; graph aggregation remains identical.

(iii) **Budget normalization.** With unit budget defined as one inference of the smallest model, the normalized budget is

$$B = f_{\text{budget}}(G, T), \qquad f_{\text{cost}}(G, T) = B \cdot f_{\text{cost}}(G_{\text{smallest}}, T).$$

## A.8 DETAILED BUDGET DEFINITION

Different model sizes and graph topologies incur substantially different computational costs: larger models introduce higher inference overhead, while denser topologies require more interactions. These differences make it challenging to establish a unified metric for budget measurement. To address this, we propose a *standardized budget definition* that enables comparability across model scales and topology complexities. For example, this framework allows us to equate the budget cost of "more sequential/parallel nodes with smaller models" to that of "fewer nodes with larger models."

Formally, let the average input and output lengths of a task be denoted by $T = (\bar{N}_p^T, \bar{N}_d^T)$. The total computational cost of a collaboration graph $G$ on task $T$ is defined as $f_{\text{cost}}(G, T)$, and the corresponding normalized budget is $B = f_{\text{budget}}(G, T)$. The cost function $f_{\text{cost}}$ can be instantiated

according to user preference to reflect different measures, such as FLOPs, wall-clock runtime, or monetary cost. To establish a common unit of comparison, we define the budget of executing one full inference with the smallest model in the pool as a single unit, i.e.,

$$f_{\text{budget}}(G_{\text{smallest}}, T) = 1,$$

where $G_{\text{smallest}}$ denotes a graph consisting of only one node of the smallest model. Consequently, the budget value of any graph $G$ is equivalent to the number of unit costs required, namely,

$$B = \frac{f_{\text{cost}}(G, T)}{f_{\text{cost}}(G_{\text{smallest}}, T)}.$$

where it means TTS graph with budget $B$ is equal to run $B$-time single-node inference.

We define the computation cost of a multi-LLM collaboration graph $G$ on a task $T = (\bar{N}_p^T, \bar{N}_d^T)$ in terms of FLOPs, which we adopt as the primary cost metric in this work. The corresponding cost function is stated in the theory below. The proof is in Appendix A.7.

*FLOPs Cost Function: For each node $v_i \in G$, the cost depends on the model size and its effective input/output lengths, leading to a quadratic dependence on the node in-degree $d(v_i)$. Summing across all nodes, the total cost can be expressed as*

$$f_{cost}(G, T) \;=\; \sum_{v_i \in \mathcal{V}} \big[\, \alpha_i \, d(v_i)^2 \;+\; \beta_i \, d(v_i) \;+\; \gamma_i \,\big],$$

*where coefficients $\alpha_i, \beta_i, \gamma_i$ capture the contributions of model dimension, depth, and average task input/output lengths. Detailed derivations of $\alpha_i, \beta_i, \gamma_i$ are provided in Appendix A.7.*

## A.9 Detailed Optimization with Joint Objective

Our optimization objective is not limited to single-performance criteria; in many cases, it is necessary to identify graph structures that satisfy composite objectives, such as achieving both low latency and high accuracy. To this end, the proposed **Agent-REINFORCE** framework incorporates diverse feedback mechanisms obtained from the `Environment` to accommodate different optimization goals. For instance, under the joint objective of low latency and high performance, we incorporate the inference time of each candidate graph as an additional feedback signal to the `Agent`. Moreover, we can explicitly provide the `Agent` with prior knowledge through instructions that describe the relationship between graph structures and latency, for example, that latency is more sensitive to the number of nodes and the graph width, thereby accelerating the search for composite-optimal graphs. All feedback, including inference time, is stored in the `Archive`, enabling the LLM to leverage historical information to assess the marginal effect of latency reduction on performance, and thus achieve a principled trade-off between efficiency and accuracy.

## A.10 Detailed Dollar Cost-based Budget

Table 5 is the API cost information for each model from Together AI and Compare Ai Models. We do not convert it in the same manner as above, as the dollar serves as a natural unit of price. Note that LLaMA-3.2 1B, Gemma-3 1B, and Gemma-1.1 2B are not quoted in Together AI or Compare Ai Models; for convenient consistency in our comparison, we adopt estimated reference values of 0.02, 0.02, and 0.06, respectively, for these models.

Table 5: Inference costs per 1M tokens for models from Together AI and Compare Ai Models.

| Model Name | Parameters | Inference Cost (per 1M tokens) |
|---|---|---|
| LLaMA-3.1 70B | 70B | $0.88 |
| LLaMA-3.1 8B | 8B | $0.18 |
| LLaMA-3.2 3B | 3B | $0.06 |
| Gemma-1.1 7B | 7B | *$0.27* |

## A.11 DETAILED REINFORCE ALGORITHM

A gradient-based algorithm can be employed to solve the optimization problem. Since the search space of collaboration graphs is prohibitively large, exhaustive enumeration of all possible configurations is infeasible. Instead, we parameterize the distribution of graphs as $\tilde{\mathbf{G}} = \mathbb{P}_{\theta,\pi,\psi}$, where $\theta$ encodes the probabilities of edge existence, $\pi$ encodes the probabilities of role assignments, and $\psi$ encodes the probabilities of model selections.

Given a budget $B$, we set the number of nodes $n$ to the maximum number of smallest models that the budget can cover. A straightforward approach to defining a parameterized distribution over DAGs with fixed $n$ nodes, edges, models, and roles is as follows. We introduce real-valued parameters: $\theta = [\theta_{ij}]$, $p_\theta(\theta_{ij}) = \sigma(\theta_{ij})$ for edge probabilities; $\pi = [\pi_1, \pi_2, \ldots, \pi_n]$ with role probabilities $p_\pi(r_i) = \mathrm{softmax}(\pi_i)$; and $\psi = [\psi_1, \psi_2, \ldots, \psi_n]$ with model probabilities $p_\psi(m_i) = \mathrm{softmax}(\psi_i)$. By iteratively refining this distribution, the algorithm progressively biases sampling toward low-loss collaboration graphs.

During training, we adopt the REINFORCE algorithm (Williams, 1992), a classical policy-gradient method that provides unbiased estimates of the utility gradient. It follows a sampling–gradient–update pipeline: candidates are sampled from the distribution, gradients are computed by evaluating on the training set, and parameters are updated via gradient ascent.

*Monte Carlo Sampling.* The probability of sampling a graph $G \sim \mathbb{P}_{\theta,\pi,\psi}$ is decomposed as

$$\mathbb{P}_{\theta,\pi,\psi} = p(\psi) \cdot p(\theta \mid \psi) \cdot p(\pi \mid \theta, \psi) = p(\psi) \cdot p(\theta \mid \psi) \cdot p(\pi \mid \theta),$$

where

$$p(\psi) = \prod_{i=1}^{n} p_\psi(\psi_i),$$

$$p(\theta \mid \psi) = \begin{cases} \prod_{i,j} p_\theta(\theta_{ij}), & \text{if the resulting graph is a DAG and } f_{\text{budget}}(G, T) \leq B, \\ 0, & \text{otherwise,} \end{cases},$$

$$p(\pi \mid \theta) = \prod_{i=1}^{n} p_\pi(\alpha^{|d(v_i)|}\pi_i), \quad \alpha \in [1, 1.1],$$

where $\alpha$ is a constant that encourages the fusion role when the in-degree of $v_i$ is high. This formulation provides a principled probabilistic parameterization of collaboration graphs, enabling efficient sampling and optimization within the REINFORCE framework.

*Gradient Estimation.* The gradient is calculated by:

$$\nabla_{\theta,\pi,\psi} \, \mathbb{E}_{G' \sim \mathbb{P}_{\theta,\pi,\psi}} \left[u_T(G')\right] = \mathbb{E}_{G' \sim \mathbb{P}_{\theta,\pi,\psi}} \left[u_T(G') \, \nabla_{\theta,\pi,\psi} \log p_{\theta,\pi,\psi}(G')\right]$$

$$\approx \frac{1}{N} \sum_{i=1}^{N} u_T(G^{(i)}) \, \nabla_{\theta,\pi,\psi} \log p_{\theta,\pi,\psi}(G^{(i)}), \tag{5}$$

where $G^{(i)}$ is the $i$-th candidate graph independently sampled from $\mathbb{P}_{\theta,\pi,\psi}$, and $N$ is the number of Monte Carlo samples.

*Parameter Updates.* The distribution parameters are then updated with gradient ascent:

$$\theta \leftarrow \theta + \tfrac{\ell}{N} \sum_{i=1}^{N} u_T(G^{(i)}) \nabla_\theta \log p_\theta(G^{(i)}),$$

$$\pi \leftarrow \pi + \tfrac{\ell}{N} \sum_{i=1}^{N} u_T(G^{(i)}) \nabla_\pi \log p_\pi(G^{(i)}), \tag{6}$$

$$\psi \leftarrow \psi + \tfrac{\ell}{N} \sum_{i=1}^{N} u_T(G^{(i)}) \nabla_\psi \log p_\psi(G^{(i)}),$$

where $\ell$ is the learning rate.

---

**Algorithm 3** REINFORCE: Optimization of the Task-Specific Multi-LLM Collaboration Graph

---

**Require:** Task $T$, training data $D_{\text{train}}$, budget $B$, learning rate $\ell$, batch size $N$
**Ensure:** Optimized distribution $\mathbb{P}_{\theta,\pi,\psi}$ and final graph $G^*$
 1: Initialize parameters $\theta$ (edge logits), $\pi$ (role logits), $\psi$ (model logits)
 2: Define distributions: $p_\theta(e_{ij}) = \sigma(\theta_{ij})$, $p_\pi(r_i) = \text{softmax}(\pi_i)$, $p_\psi(M_i) = \text{softmax}(\psi_i)$
 3: **while** stopping criterion is not met **do**
 4: $\quad$ $\mathcal{B} \leftarrow \emptyset$ $\hfill$ ▷ initialize mini-batch of sampled graphs
 5: $\quad$ **for** $i = 1$ to $N$ **do**
 6: $\quad\quad$ $G^{(i)} \sim \mathbb{P}_{\theta,\pi,\psi}$ $\hfill$ ▷ sample edges, roles, and models
 7: $\quad\quad$ **if** $f_{\text{budget}}(G^{(i)}, T) > B$ **then**
 8: $\quad\quad\quad$ **continue** $\hfill$ ▷ reject graph if budget exceeded
 9: $\quad\quad$ **end if**
10: $\quad\quad$ $u_i \leftarrow u_T(G^{(i)}, D_{\text{train}})$ $\hfill$ ▷ evaluate utility
11: $\quad\quad$ $\mathcal{B} \leftarrow \mathcal{B} \cup \{(G^{(i)}, u_i)\}$
12: $\quad$ **end for**
13: $\quad$ $g_\theta \leftarrow \frac{1}{|\mathcal{B}|} \sum_{(G,u)\in\mathcal{B}} u \, \nabla_\theta \log p_\theta(G)$
14: $\quad$ $g_\pi \leftarrow \frac{1}{|\mathcal{B}|} \sum_{(G,u)\in\mathcal{B}} u \, \nabla_\pi \log p_\pi(G)$
15: $\quad$ $g_\psi \leftarrow \frac{1}{|\mathcal{B}|} \sum_{(G,u)\in\mathcal{B}} u \, \nabla_\psi \log p_\psi(G)$
16: $\quad$ $\theta \leftarrow \theta + \ell \, g_\theta; \quad \pi \leftarrow \pi + \ell \, g_\pi; \quad \psi \leftarrow \psi + \ell \, g_\psi$ $\hfill$ ▷ gradient ascent updates
17: **end while**
18: Construct $G^*$ by MAP decoding: include edge $e_{ij}$ if $p_\theta(e_{ij}) \geq \tau_e$; set role $r_i \leftarrow \arg\max_r p_\pi(r_i{=}r)$; set model $M_i \leftarrow \arg\max_m p_\psi(M_i{=}m)$ $\hfill$ ▷ deterministic final graph
19: Ensure $f_{\text{budget}}(G^*, T) \leq B$ (greedy prune if needed)
20: **return** $\mathbb{P}_{\theta,\pi,\psi}$ and $G^*$

---

*Optimization loop.* REINFORCE alternates between three phases: (i) *sampling*, where candidate graphs $G^{(i)}$ are drawn from the current distribution; (ii) *evaluation*, where utilities $u_T(G^{(i)})$ are computed on the training set; and (iii) *update*, where parameters $\theta, \pi, \psi$ are refined by gradient ascent. This process repeats until convergence or when the budget is exhausted.

*Final graph selection.* After optimization, the learned distribution $\mathbb{P}_{\theta,\pi,\psi}$ is used to construct a deterministic collaboration graph $G^*$. Specifically, we decode by maximum a posteriori (MAP): edges are included if $p_\theta(e_{ij}) \geq \tau_e$, roles are assigned as $r_i = \arg\max_r p_\pi(r_i{=}r)$, and models are chosen as $M_i = \arg\max_m p_\psi(M_i{=}m)$. The final graph is pruned if necessary to ensure $f_{\text{budget}}(G^*, T) \leq B$. The complete optimization procedure is summarized in Algorithm 3.

## A.12 Prompt Design in Agent-REINFORCE

We design structured prompts to guide the LLM search agent in initializing and updating the collaboration graph. Each prompt provides task context, distilled insights, and design constraints to support systematic reasoning and planning. For model family and size initialization, the agent ranks candidate families and sizes under budget constraints, guided by single-model performance and preliminary evaluations. This establishes a principled starting point for subsequent exploration. For model instance count initialization, the agent specifies concrete model combinations with family, size, and instance counts. These candidates are then tested in the environment, and the feedback highlights the most promising allocations. For graph updates, the agent leverages Insight 2, Insight 3, and feedback from the previous round to refine edge distributions, adjust connectivity, and balance budget allocation, thereby improving the overall structure and moving toward compute-optimal performance.

---

**LLM Prompt for Model Family and Size Preference Initialization**

**Your current task is model family and size initialization:** you must provide the model family and size preferences for a test-time collaboration graph that will later be optimized into a DAG. An edge indicates that the previous model's output is the next agent's input.

=============== **TASK** ================
1. Examine the candidate model combinations listed at the end of this message.
2. Return a JSON dictionary of model family and size ranking.
No extra text, explanations, or formatting—just the dictionary.

============== **INSIGHTS** ==============
(1) Different tasks exhibit a clear preference for specific model combinations. Under budget constraints, it is necessary to identify the preferred model family and model size for each task.

============== **DATA** ==================
Single-model accuracy on {task} (higher is better):
{model_profile} or {pre_test_accuracy}
Random-graph pre-experiment results (including small models running once or twice and large models running once):
{combinations_accuracy}

============== **CANDIDATES** =============
Choose only one from this list (each already fits the budget):
{model_combinations}

=========================================
Respond with the dictionary only. Example format:

---

**LLM Prompt for Model Instance Counts Initialization**

**Your current task is model instance count initialization:** you must provide the model instances for a test-time collaboration graph that will later be optimized into a DAG. We will test them in the Environment and select the best one according to the feedback. An edge indicates that the previous model's output is the next agent's input.

=============== **TASK** ================
1. Examine the model family and size preferences listed at the end of this message.
2. Return a JSON dictionary of model combinations with model family, size, and instance counts.
No extra text, explanations, or formatting—just the dictionary.

============== **INSIGHTS** ==============
(1) Different tasks exhibit a clear preference for specific model combinations. Under budget constraints, it is necessary to identify the preferred model family and size for each task.

============== **PREFERENCE** =============
Model family and size preferences:
{model_family_size_preference}

============== **DATA** ==================
Single-model accuracy on {task} (higher is better):
{model_profile} or {pre_test_accuracy}

=========================================
Respond with the dictionary only. Example format:

**LLM Prompt for Graph Updates**

**You are a professional Multi-LLM system optimizer.** Your task is an iterative self-RL refinement of a multi-LLM system that solves the {task} dataset.

**TASK CONTEXT**
• A Multi-LLM system is represented as a directed acyclic graph (DAG). Each node = one language-model agent. Each directed edge = "the source agent's output is appended to the destination agent's context".
• For the current budget, we have a fixed model-selection requirement: {model_selection}
• You will see the last-round graph, its batch accuracy, and the full table of edge-selection probabilities.
• Your job: propose the next-round graph and the updated probability table, applying RL-style probability nudges.
• The graph you receive in this iteration has been expanded outward from the FinalDecision node, gradually increasing in both depth and breadth. The edge probabilities start with all edge probabilities set to zero, and through multiple sampling rounds, probabilities are raised only for edges that prove useful.

**HISTORICAL SNAPSHOT**
Last-round accuracy ({task}-dev batch): {accuracy}
Last-round graph: {prev_graph}
Last-round edge-probabilities: {edge_probs}

**OPTIMIZATION RULES**
R-1 Model counts must exactly match model selection after you assign models to all nodes.
R-2 A node's role is either "assistant" (generates a new answer) or "fuser" (reviews & picks the best).
R-3 Increase an edge probability only if it was sampled in the last-round graph AND proved useful. Always start expansion from FinalDecision's incoming edges, then its parents' incoming edges, and so on. Increase edges used by high-accuracy graphs, decrease edges from poor graphs.
R-4 Keep the graph acyclic; avoid too much in-degree to prevent context explosion; avoid very deep chains to prevent "answer corruption".

**DATA AND INSIGHT**
• Model accuracy on {task} (single-agent): {model_profile}
• The optimal depth is conditioned by current width, and vice-versa: wider graphs shift the depth sweet-spot downward, while deeper graphs reduce the optimal width.
• You should expand the architecture outward from the FinalDecision node, gradually adding depth and width.
• Different tasks favor different graph topologies; optimize toward the topology style that this task prefers.

**WHAT TO RETURN**
• graph — the next-round DAG, same schema as last-round graph.
• edge probs — the updated probability table, same schema and order as last-round edge-probabilities.

**Example output format (do NOT add comments):**
Graph: {graph_example}
Edge-probabilities: {node_example}

Now think step-by-step with the rules and insights above, and return the Graph and Edge-probabilities two blocks only.

## A.13 BASELINES

We compare three baseline categories: LLM-based (MaaO (Guo et al., 2024) and TextGrad (Yuk-sekgonul et al., 2024)), gradient-based (GPTSwarm (Zhuge et al., 2024)), and traditional methods (Bayesian Optimization (Shahriari et al., 2015) and Random Search). Then, we detail their adaptation.

**TextGrad (Yuksekgonul et al., 2024)** performs automatic "differentiation" through text, where an LLM generates a natural language "gradient" that guides updates to optimizable variables based on predictions and loss values. In the context of compute-optimal collaboration graph optimization for TTS, the probabilistic graph serves as the optimizable variable. Candidate graphs are sampled from the current distribution and evaluated on a batch of training data to compute the loss; the LLM then provides textual guidelines indicating how the graph should be refined given the observed loss and inputs. This process is repeated iteratively until convergence or a predefined stopping criterion is met. During initialization, TextGrad selects the maximal model combination that encompasses all potential candidates (i.e., allocating nodes to every feasible mixture of available models within the budget). Compared with our method, TextGrad lacks task-specific initialization and test-time scaling knowledge, making it a less efficient and less effective baseline.

**MaaO (Guo et al., 2024).** is a hybrid approach that integrates gradient-based optimization with LLM-guided optimization, leveraging the complementary strengths of both. Gradient-based methods provide precise directional updates in the parameter space but are prone to local optima, while LLM optimizers offer high-level heuristic guidance yet often lack stability. To address this, MaaO alternates between the two optimization strategies. In our problem setting of optimizing probabilistic graphs, we adopt REINFORCE to compute numerical gradients and use an LLM to generate textual updates, alternating between them during training. Concretely, the probabilistic graph is first initialized with a uniform distribution (same as described above), from which candidate graphs are sampled and evaluated on a training batch to compute predictions and loss values. Gradients derived from the loss are then used to update the probabilistic graph (see Appendix A.11). Subsequently, new candidates are sampled, and their losses are used by the LLM to provide textual updates on how the graph should be modified. This alternating process of gradient updates and LLM guidance continues until convergence or a stopping criterion is met.

**GPTSwarm (Zhuge et al., 2024)** generalizes LLM-based agent architecture search into a computational graph and optimizes it using gradient-based REINFORCE. In our problem setting, we adapt this approach as follows: a probabilistic graph is first initialized, from which candidate graphs are sampled and evaluated on a batch of training data to compute predictions and loss values. The loss gradients are then used to update the probabilistic graph, and this process is iterated until a stopping criterion is reached. The detailed REINFORCE optimization procedure is in Appendix A.11. However, as a purely gradient-based approach, GPTSwarm is relatively inefficient, as each update makes only incremental progress, and the method is susceptible to convergence at suboptimal local minima, thereby limiting both convergence speed and global search capability.

**Bayesian Optimization (BO) (Shahriari et al., 2015)** is a model-based framework for black-box optimization and has been widely applied to hyperparameter tuning. For optimizing collaboration graphs in test-time scaling, the graph is parameterized by $\theta, \pi, \psi$, from which a concrete graph $G$ is sampled and evaluated on a training batch to obtain its performance $f(G)$. Accordingly, BO treats $\theta, \pi, \psi$ as input variables, with the objective function defined as $F(\theta, \pi, \psi) = \mathbb{E}_{G \sim \mathcal{P}_{\theta, \pi, \psi}}[f(G)]$. Specifically, BO constructs a surrogate model, such as a Gaussian process, to approximate $F(\theta, \pi, \psi)$, and employs an acquisition function (e.g., Expected Improvement, EI) to guide the selection of promising candidates. Each selected $(\theta, \pi, \psi)$ is evaluated by sampling multiple graphs to estimate average performance. Under budget constraints, the cost function $f_{\text{budget}}(G)$ can be incorporated via constrained acquisition (e.g., constrained EI). This iterative process of surrogate modeling, candidate selection, and evaluation continues until a stopping criterion is reached, at which point BO returns the optimal parameter set $(\theta^\star, \pi^\star, \psi^\star)$ and its corresponding high-performing probabilistic graph.

**Random Search** is a simple but widely adopted baseline in hyperparameter optimization. For compute-optimal collaboration graph search in test-time scaling, it generates candidate graphs uniformly at random under the budget constraint, without leveraging prior knowledge or performance history. While its simplicity makes it robust to irregular or non-smooth search landscapes and occasionally capable of identifying strong candidates, the absence of guidance typically leads to inferior search efficiency and performance compared with more structured or informed methods.

## A.14 CONVERGENCE AND EFFICIENCY ON MATH DATASET

As shown in Fig. 9, our method achieves the best accuracy and fastest convergence via strong initialization and guided by empirical insights. TextGrad tends to overuse the budget and slows down, while GPTSwarm/MaaO converges quickly but gets stuck in local optima.

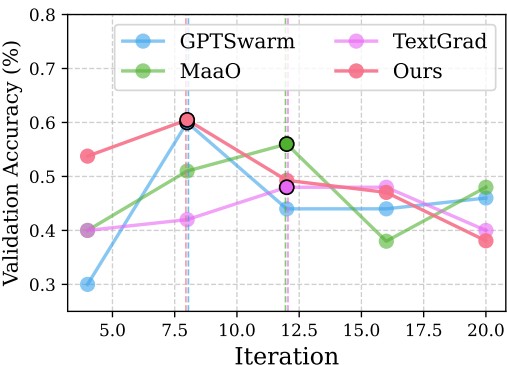

Figure 9: Training trajectories on MATH across LLM-based methods over 20 iterations. X-axis: iteration; Y-axis: the validation accuracy.

## A.15 A QUALITATIVE EXAMPLE OF TTS GRAPH AND ITS OUTPUT

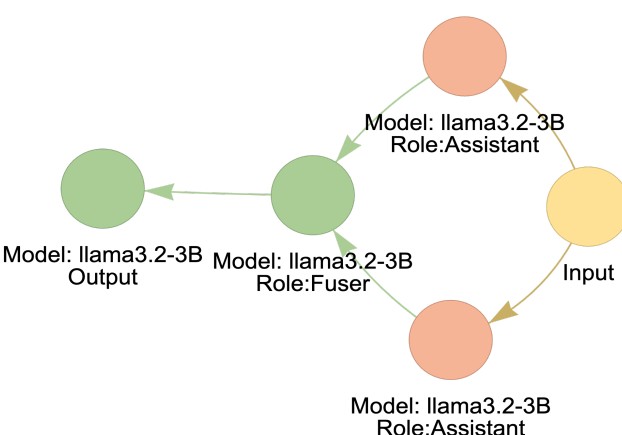

Figure 10: A qualitative example of TTS graph.

We present a quantitative example in Fig. 10 using an instance from the MATH dataset. The graph contains five nodes: one input node, one output node, two Assistant nodes, and one Fuser node. The topology consists of two Assistant nodes feeding into the Fuser node, which then connects to the output node. We provide the prompts and outputs for each node in the text boxes below. As shown, one Assistant node produces the correct answer while the other produces an incorrect one; the Fuser successfully selects the correct output. The output node receives input from a single node and therefore simply copies that node's output (details omitted for brevity).

---

**Input Node**

**System Message:** You are an expert mathematician. Help the user to solve this problem. Answer the following mathematics question. Provide your reasoning by showing your work before your answer. At the end of your response, output your final answer in the format: 'The answer is: [answer]'. You must provide the separator 'The answer is: ' before your final answer. Make sure to put the answer (and only the answer) inside \boxed{}.

**User Message:** If $f(x) = \dfrac{a}{x+2}$, solve for the value of $a$ so that $f(0) = f^{-1}(3a)$.

---

---

**Assistant 1 Node Output**

To find the inverse function $f^{-1}(x)$, we swap $x$ and $y$ in $f(x) = \frac{a}{x+2}$ and solve: $x = \frac{a}{y+2} \Rightarrow x(y+2) = a \Rightarrow xy + 2x = a \Rightarrow y = \frac{a-2x}{x}$, so $f^{-1}(x) = \frac{a-2x}{x}$.

We are given $f(0) = f^{-1}(3a)$. Compute $f(0) = \frac{a}{2}$. Next, $f^{-1}(3a) = \frac{a-6a}{3a} = -\frac{5}{3}$.

Set them equal: $\frac{a}{2} = -\frac{5}{3}$. Multiply by 6: $3a = -10$, so $a = -\frac{10}{3}$.

The answer is: \boxed{$-\frac{10}{3}$}.

---

**Assistant 2 Node Output**

To find the inverse function $f^{-1}(x)$, we swap $x$ and $y$ in $y = \frac{a}{x+2}$. After swapping, we get $x = \frac{a}{y+2}$, which simplifies to $xy + 2x = a$, then $xy = a - 2x$, and finally $y = \frac{a-2x}{x}$. Thus, $f^{-1}(x) = \frac{a-2x}{x}$.

We now solve for $a$ such that $f(0) = f^{-1}(3a)$. Since $f(x) = \frac{a}{x+2}$, we have $f(0) = \frac{a}{2}$. Next, compute $f^{-1}(3a) = \frac{a-6a}{3a} = -\frac{5a}{3a} = -\frac{5}{3}$.

Setting them equal: $\frac{a}{2} = -\frac{5}{3}$. Cancelling $a \neq 0$ gives $\frac{1}{2} = -\frac{5}{3}$, which is impossible, so no real value of $a$ satisfies the equation.

---

**Fuser Node Prompt and Output**

**System Message:**

You are a fusion agent in a multi-agent system. Your role is to review, condense, and synthesize the reasoning and answers produced by previous agents.

**User Message:**

According to the previous agents' answers for the problem:

*Question:* If $f(x) = \frac{a}{x+2}$, solve for $a$ so that $f(0) = f^{-1}(3a)$.

As the fusion agent, synthesize a high-quality final answer by extracting correct reasoning, removing redundancy, ensuring mathematical rigor, and producing a clear justification. The final answer must be wrapped in \boxed{}.

**Previous Answers:**

**1. Previous Agent (Version 1):**

Correctly computed $f^{-1}(x) = \frac{a-2x}{x}$, $f(0) = \frac{a}{2}$, and $f^{-1}(3a) = -\frac{5}{3}$.

Correctly solved $\frac{a}{2} = -\frac{5}{3} \Rightarrow 3a = -10 \Rightarrow a = -\frac{10}{3}$.

**2. Previous Agent (Version 2):**

Derived $f^{-1}(x)$ by swapping $x, y$ in $y = \frac{a}{x+2}$, leading to $x = \frac{a}{y+2}$, then $xy + 2x = a$, $xy = a - 2x$, and $y = \frac{a-2x}{x}$. Thus $f^{-1}(x) = \frac{a-2x}{x}$.

Computed $f(0) = \frac{a}{2}$ and $f^{-1}(3a) = \frac{a-6a}{3a} = -\frac{5}{3}$.

Equating gave $\frac{a}{2} = -\frac{5}{3}$, but incorrectly concluded no solution due to cancelling $a$ incorrectly.

**Fuser Output:**

**Synthesis of the Solution.** Both previous agents correctly derived the inverse function $f^{-1}(x) = \frac{a-2x}{x}$. Agent 2 provided the cleanest evaluation of $f(0)$ and $f^{-1}(3a)$.

**Step 1:** $f(0) = \frac{a}{2}$.

**Step 2:** $f^{-1}(3a) = -\frac{5}{3}$.

**Step 3:** Set $\frac{a}{2} = -\frac{5}{3}$. Multiply both sides by 6 to clear denominators: $3a = -10$, so $a = -\frac{10}{3}$.

**Step 4:** Substitution verifies correctness.

**Final Answer:** \boxed{$-\frac{10}{3}$}.

---

## A.16 DETAILED RELATED WORK

**Test-time Scaling and Compute-optimal Strategy.** Inspired by the human tendency to allocate additional cognitive effort for deeper and more deliberate reasoning, recent studies have proposed distributing extra computational resources during inference to improve model performance on various tasks (Wei et al., 2022; Wang et al., 2023). In parallel, other works (Brown et al., 2024; Wu et al., 2025) have observed that increasing inference-time computation follows a scaling law analogous to that of training, where additional computation consistently enhances task performance. This phenomenon is commonly referred to as *Test-Time Scaling (TTS)*. Existing TTS techniques can be broadly categorized into two paradigms: *sequential scaling* and *parallel scaling*. In sequential scaling, the model enhances its reasoning ability by progressively extending a reasoning chain. A common approach is *self-refinement*, in which the model first generates an initial response and then iteratively revises it based on self-assessment (Madaan et al., 2023; Gou et al., 2024; Snell et al., 2025; Chen et al., 2024c; 2025). Because this strategy depends heavily on the quality of the initial

output, it tends to be more effective on relatively simple tasks (Snell et al., 2025). By contrast, parallel scaling improves inference by generating multiple independent candidate solutions simultaneously and aggregating them into a final answer. Representative aggregation strategies include *majority voting* (Liu et al., 2025b; Wang et al., 2023), which selects the most frequent output among $N$ candidates, and *Best-of-N* (Brown et al., 2024; Sun et al., 2024; Gui et al., 2024), which samples $N$ solutions and uses a verifier to select the best one (Setlur et al., 2025). Other approaches employ LLMs themselves as fusers to integrate multiple candidates into a single output, thereby providing stronger generalization and flexibility (Jiang et al., 2023; Li et al., 2025b; Saad-Falcon et al., 2024). Despite these successes, both paradigms exhibit limitations. Sequential scaling suffers from poor scalability, as extending the reasoning chain increases the risk of corrupting previously correct intermediate results (Zeng et al., 2025). Parallel scaling, while improving diversity, often lacks the depth of reasoning required for more complex tasks (Misaki et al., 2025). To address these issues, hybrid approaches have been explored. For instance, Snell et al. (2025) propose adaptively switching between sequential and parallel scaling depending on task difficulty, using sequential scaling for simpler tasks and parallel scaling for more complex ones. Other methods leverage tree-structured search to combine the two paradigms at the step or output level, employing process-level reward models to expand top-$K$ intermediate steps and refine them further. Typical examples include beam search (Yu et al., 2024; Xie et al., 2023) and Monte Carlo Tree Search (MCTS) (Wu et al., 2025; Snell et al., 2025; Hao et al., 2023; Wan et al., 2024; Chen et al., 2024a; Zhang et al., 2023). Nevertheless, most existing hybrid methods assume a *fixed inference structure* (e.g., fixed width or depth), limiting their flexibility. Recent studies have begun to relax these assumptions. For example, *Adaptive Parallel Reasoning* (Pan et al., 2025) dynamically switches between sequential and parallel computation using `spawn` and `join` operations, while *Adaptive Branching MCTS* unifies both paradigms within a tree-search framework, deciding at each node whether to parallelize candidate generation or continue sequential refinement. In addition, prior work has noted that sampling across multiple models naturally falls within the scope of test-time scaling, since ensembles improve diversity and output quality (Zhang et al., 2025b; Ashiga et al., 2025; Jiang et al., 2023), yet this dimension remains underexplored in test-time scaling.

The configuration of allocating computation at inference time is central to the effectiveness of test-time scaling (TTS), giving rise to the *compute-optimal test-time scaling strategy*. A growing body of work (Brown et al., 2024; Wu et al., 2025; Liu et al., 2025a; Yue et al., 2025; Snell et al., 2025; Wang et al., 2025a) highlights that model size and scaling configuration must be carefully balanced: in certain scenarios, smaller models can achieve superior accuracy compared to large models when constrained by the same compute budget. This line of research explores both model selection, deciding when to employ small versus large models, and method selection, choosing between alternative scaling paradigms to maximize utility. For instance, Snell et al. (2025) show that the optimal scaling strategy varies with task difficulty: moderately challenging tasks favor parallel exploration with small models, whereas simpler tasks are better addressed through sequential refinement with large models. They further introduce a difficulty predictor to adaptively switch strategies. Other studies extend these ideas in different directions: Liu et al. (2025a) emphasize the sensitivity of scaling strategies to reward design, Yue et al. (2025) develop a linear model to capture key determinants of scaling within retrieval-augmented generation (RAG), and Wu et al. (2025) propose Reward Balanced Search (REBASE), a tree-search algorithm that achieves a Pareto-efficient balance between accuracy and inference cost through weighted voting. Despite these advances, existing approaches remain limited to fixed inference structures, overlooking the richer TTS patterns that arise in general graph topologies. Motivated by these gaps, we address a novel problem: unifying test-time scaling under a graph-based framework that incorporates heterogeneous model combinations, and searching for the compute-optimal collaboration graph.

**Multi-agent Collaboration Graph.** With the emergence of LLMs and the rapid development of LLM-based agents (Cohen et al., 2023; Zhuge et al., 2024), researchers have increasingly recognized that interactions among multiple agents can be naturally represented from a graph-based perspective (Chen et al., 2024b; Zhuge et al., 2024; Qian et al., 2025; Liu et al., 2024c). Graphs provide a principled abstraction for capturing communication patterns, role assignments, and coordination strategies in multi-agent systems, making them well-suited for reasoning about collaborative intelligence. Recent systems such as G-Designer (Zhang et al., 2025a), ARG-Designer (Li et al., 2025a), Heterogeneous Swarms (Feng et al., 2025), DyLAN (Liu et al., 2024c), AgentNet (Yang et al., 2025), GPTSwarm (Zhuge et al., 2024), and MacNet (Qian et al., 2025) have explicitly employed graph

structures to organize and optimize multi-agent interactions. These approaches primarily focus on structural optimization over a predefined set of agents, selecting the structure that maximizes task performance, which can be partially applied to our problem setting. However, they overlook the distinctive patterns of test-time scaling, resulting in inefficient architecture search.

**LLMs for Optimization**  Optimization is fundamental to computational models and is often customized for individual tasks to address the challenges of complex decision spaces and performance landscapes. Large Language Models (LLMs), with their rich prior knowledge and reasoning capabilities, have opened new avenues for solving practical optimization problems (Zhang et al., 2025c; Guo et al., 2024). Existing research primarily employs LLMs in two paradigms: as black-box optimizers and in conjunction with gradient-based white-box optimization. The distinction lies in whether gradient information is available. In the black-box setting, LLMs are used to generate candidate solutions and iteratively refine them by leveraging their planning ability and extensive machine learning knowledge. Prior work has demonstrated the effectiveness of this approach in small-scale mathematical optimization (Yang et al., 2024; Zhang et al., 2024b; Huang et al., 2025), hyperparameter tuning (Liu et al., 2024a;b), and neural architecture search (Zheng et al., 2023; Nasir et al., 2024; Ji et al., 2025). For instance, OPRO (Yang et al., 2024) proposed "optimization by prompting," where tasks are described in natural language and LLMs iteratively generate new solutions based on meta-prompts and prior evaluations. AgentHPO (Liu et al., 2024a) empowers LLMs to autonomously search hyperparameter configurations by processing task descriptions, conducting experiments, and refining search quality from accumulated trials. GENIUS (Zheng et al., 2023) explored the potential of GPT-4 for neural architecture search, employing its generative ability as a black-box optimizer to efficiently navigate the search space and refine promising architectures. LLMs are particularly valuable during initialization, as they can generate high-quality solutions that embed prior knowledge, narrowing the search space and establishing a stronger foundation for subsequent iterations. This capability has also been applied to NAS initialization (Jawahar et al., 2024), genetic algorithms in bioengineering (Nana Teukam et al., 2025), and financial planning (De Zarzà et al., 2023).

These studies demonstrate that LLMs can serve as general-purpose black-box optimizers. However, when gradient information is available—typically in data-rich scenarios—black-box optimization becomes inefficient, as each candidate must be evaluated on the full training set, leading to prohibitive search costs. To address this, recent work has combined gradient-based optimization with LLM-guided search to exploit their complementary strengths (Guo et al., 2024; Yuksekgonul et al., 2024). For example, MaaO (Guo et al., 2024) interleaves gradient-based training with LLM-guided optimization, integrating the data efficiency and precise updates of gradient methods with the exploratory diversity of LLMs. TextGrad (Yuksekgonul et al., 2024) generalizes this idea by transforming AI systems into computational graphs and using LLMs to generate textual updates that serve as a form of backpropagation. This framework provides natural language critiques of system components, such as neurons, prompts, molecules, or code segments, and guides their updates. Building on this line of work, we extend the complementary use of LLMs and gradient methods to compute-optimal test-time scaling by optimizing a gradient-available probabilistic graph. This approach enables us to combine the data efficiency of gradient-based optimization with the semantic task-awareness of LLMs, particularly for critical initialization and text-form parameter updates, thereby improving both search effectiveness and efficiency.

## B  LLM USAGE

Large Language Models (LLMs) were used solely for language refinement, including rephrasing, grammar checking, and improving readability. They were not involved in ideation, methodology, experiments, or data analysis. All research concepts and results are the authors' own, and the authors take full responsibility for the manuscript, ensuring that LLM-assisted text complies with ethical standards and avoids plagiarism or misconduct.

