# OpenReview forum: "Agent-REINFORCE: Searching Compute-Optimal Multi-LLM Collaboration Graph for Test-Time Scaling"
_ICLR.cc/2026/Conference — Submitted to ICLR 2026_

### Official Review · Reviewer_UoVT · 2025-10-26

**Soundness:** 3
**Presentation:** 3
**Contribution:** 3
**Rating:** 6
**Confidence:** 4

**Summary:**

This work proposes Agent-Reinforce, an algorithm to optimize a computation graph of multiple language models for collaborative problem solving. Empirical results show its promise against various baselins.

**Strengths:**

+ multi-llm collaboration is an important direction
+ formulating the problem as a graph optimization problem is nice

**Weaknesses:**

- The paper argues that many things are fixed in previous works (graph structure, etc.), while this work makes them flexible and optimizable. One thing becomes fixed in this work though: the role of model/agents. In section 3, the authors hand-crafted two roles: assistant and fuser, with their specific prompts and configurations. 1) Do you forsee any other potential role? 2) Is it possible to make roles a bit more flexible than hand-crafting prompts and manually designing roles? 3) If a new role or two is added, how would the system cope?

- How is "Theorem 1", line 186, a theorem? It's a conclusion about the cost of the computational graphs under a set of hyperparameters, not a theorem that was "proved" by anything?

- There does not seem to be any qualitative example (specific generated texts by the models in the computation graph), main paper or appendix. Maybe I missed it.

- The word "(test-time) scaling" was mentioned many times as a motivation for this approach. It would be nice to have some sort of scaling diagram, that by allowing more more models/larger graph sizes the performance goes up to some extent.

**Questions:**

please see above

---

> ### Author Response · Authors · 2025-11-24
>
> We thank the reviewer for recognizing the importance of the research problem and for the valuable reviews. We provide our point-by-point responses below.
>
> > 1) Do you foresee any other potential role? 2) Is it possible to make roles a bit more flexible than hand-crafting prompts and manually designing roles? 3) If a new role or two is added, how would the system cope?
>
> We indeed use fixed role definitions in our framework. This is because our method abstracts the Test-Time Scaling setting from two canonical strategies, parallel best-of-N sampling and sequential self-refinement, both of which naturally correspond to only two essential roles: Fuser and Assistant.
>
> Regarding (1) the possibility of additional roles, we agree that more role types may exist. For example, in [1], beyond the Fuser and Generator (which corresponds to our Assistant), the system also includes roles such as Critic, Ranker, Verifier, Unit Test Generator, and Unit Test Evaluator. These roles could also be incorporated into our setting when more diverse reasoning strategies are considered for enhancing TTS performance.
>
> As for (2) whether role design can be made more flexible, the answer is yes, although our current method does not explore this direction. A more adaptive design could leverage OPRO [2], allowing each node’s role prompt to be optimized automatically through LLM-generated feedback. An even more flexible alternative is to remove predefined roles entirely: new nodes could learn their roles during training. As a node repeatedly collaborates effectively with certain existing roles on high-performing tasks, it could gradually develop an emergent, adaptive role.
>
> Finally, for (3), our framework is compatible with expanding the role set. In optimization, each node is associated with a role assignment vector. Currently, with two roles, this is a two-dimensional vector. To add more roles, we only need to increase the dimensionality of this vector, making role expansion straightforward in our architecture.
>
> [1] Saad-Falcon, Jon, et al. "Archon: An architecture search framework for inference-time techniques." arXiv preprint arXiv:2409.15254 (2024).
>
> [2] Yang, Chengrun, et al. "Large language models as optimizers." The Twelfth International Conference on Learning Representations. 2023.
>
> > How is "Theorem 1", line 186, a theorem?
>
> We agree with the reviewer’s comment; the original label was used mainly for formatting purposes. Since the statement does not require a formal proof and can be directly derived from the existing basic math operations, the term “theorem” is indeed inappropriate. To address this issue, we have replaced “theorem” with “proposition.” The term “proposition” is the standard choice for mathematically derived properties that are informative but not intended as major theoretical contributions.
>
> > There does not seem to be any qualitative example
>
> We thank the reviewer for pointing out this omission. You are correct that the previous version did not include a qualitative example. In the updated PDF, we have added such an example in Appendix A.15. Specifically, we provide a simple TTS graph consisting of five nodes and illustrate its behavior on a sample problem from the MATH dataset. For each node, we include both the input prompt and the generated output. As shown in the example, the Fuser node successfully selects the correct response from the two Assistant nodes (one producing a correct answer and the other producing an incorrect one). This qualitative illustration clarifies how the proposed approach integrates and refines multiple model outputs during test-time computation.
>
> > It would be nice to have some sort of scaling diagram, that by allowing more models/larger graph sizes the performance goes up to some extent.
>
> We provide the following results showing that as the budget increases, allowing larger graph sizes and more model choices, performance improves and then plateaus once the optimal budget is reached. Beyond this point, additional compute does not yield further gains, indicating saturation behavior.
>
> | budget range (on MATH) | ACC |
> | ------------------- | ----- |
> | 1–10                | 33    |
> | 10–18               | 40     |
> | 18–29               | 47    |
> | 29–51               | 50    |
> | 51–80               | 56    |
> | >80                 | 56    |
>
>
> We hope these responses address the reviewer’s concerns and are happy to clarify further if needed.

---

> > ### Comment · Reviewer_UoVT · 2025-11-25
> >
> > I would like to thank the authors for their detailed response.

---

> ### Author Response · Authors · 2025-11-27
>
> We thank the reviewer for the valuable feedback again. Please let us know if you have any additional concerns. If not, we kindly request your consideration for increasing the rating score.

---

### Official Review · Reviewer_yy82 · 2025-10-28

**Soundness:** 3
**Presentation:** 4
**Contribution:** 3
**Rating:** 6
**Confidence:** 3

**Summary:**

The work study the problem of searching for compute-optimal model combinations and architectures in test-time scaling under a fixed budget. It introduces Agent-REINFORCE, an LLM-agent framework that conducts budget-aware, feedback-driven search on collaboration graphs. Experiments show that the proposed algorithm outperforms baselines in both search efficiency and performance.

**Strengths:**

- Overall, I really like the proposed method. Using agent to search for model combinations and architectures in TTS is an interesting and well-motivated idea.

- The work explicitly incorporates budget constraints, either FLOPs or dollar cost, which provides a realistic perspective on multi-LLM collaboration in real deployment scenarios.

- The experimental setup is thorough, and the results are well-supported with clear justification, detailed analysis, and informative ablations.

- The paper is exceptionally well-presented. The writing is clear, figures and plots are informative, and the algorithmic components are introduced with clarity. I gave presentation score of 4.

**Weaknesses:**

The proposed method demonstrates strong performance in both accuracy and efficiency according to Table 1, substantially outperforming the baselines. That said, I have two concerns:

- Table 1 omits results for a single-model baseline, which makes it difficult to gauge how much of the gain is truly due to multi-LLM collaboration rather than simply using a stronger model. For instance, according to Figure 3(c), a 3B Mix model can achieve 54 on MATH, whereas a single LLaMA-8B reaches 64 on MMLU, which exceeds both the baseline and proposed method in terms of both accuracy and efficiency.

- The cost analysis appears to exclude the overhead associated with the search agent (i.e., DeepSeek‑R1), which is expensive.

**Questions:**

- what if we don't pre-assign roles?

---

> ### Author Response · Authors · 2025-11-22
>
> We sincerely appreciate the reviewer’s positive evaluation of our method and the constructive feedback provided. We respond to each of the concerns in detail below.
>
> > Table 1 omits results for a single-model baseline, which makes it difficult to gauge how much of the gain is truly due to multi-LLM collaboration rather than simply using a stronger model. For instance, according to Figure 3(c), a 3B Mix model can achieve 54 on MATH, whereas a single LLaMA-8B reaches 64 on MMLU, which exceeds both the baseline and proposed method in terms of both accuracy and efficiency.
>
>
> We provide the single-model baseline results for a fair comparison. Specifically, we use the strongest individual model (Llama3-8B) as the single-model baseline because it consistently yields the best performance among standalone models. The results under a budget of 80 (now also included in the updated PDF) are shown below:
>
> |                          | MATH | MMLU | HumanEval | Avg   |
> | ------------------------ | ---- | ---- | --------- | ----- |
> | Single-model (Llama3-8B) | 49   | 64   | 60        | 58|
> | BO | 42| 36.8 |33| 38|
> | TextGrad | 41 | 40  | 42  | 43 |
> | Ours                     | 56   | 61.5 | 73        | 64 |
>
> These results show that prior baselines often fail to discover combinations that outperform simply using the largest model. This is largely because these methods tend to overspend the available budget, which in turn degrades model quality.
>
> After conducting a significance analysis, our MMLU result was updated from 54→61.5 after running 10 times. Although our method achieves a strong average performance, it is true that on MMLU, directly using the 8B model yields the highest score under a budget of 80. This is expected: the optimal allocation for MMLU at this budget is a single 8B model. Our method frequently discovers combinations such as “8B + 3B,” which slightly reduces accuracy than the optima, though it occasionally also identifies the pure 8B solution. In contrast, baselines tend to select multiple small models or low-quality combinations such as “8B + too many small models,” which indicates that they are less effective at avoiding unnecessary budget usage.
>
> It is important to note that the largest model is not always the best choice. On datasets such as MATH and HumanEval, our method discovers model-mix configurations that outperform the single 8B baseline. For example, prior results show that a 3B-Mix achieves 54 on MATH, close to the best-known performance, and our searched model mixtures attain better results. This demonstrates that our method is capable of finding high-performing small-model combinations rather than relying solely on scaling up model size.
>
> > The cost analysis appears to exclude the overhead associated with the search agent (i.e., DeepSeek‑R1), which is expensive.
>
> We report the inference cost (both time and dollar cost) on the MATH dataset under a budget of 80, separating the initial stage and the search iteration stage, as in the following table. The overall time and dollar cost of our method remain small because the training set we use is small.
>
> In terms of time cost, the search stage involves up to 30 iterations, corresponding to 30 batches, and is divided into two parts: trial generation by the LLM agent and trial execution in the Environment. The trial-generation stage dominates the total search time, accounting for approximately 526/900 ≈ 58% of the total. Execution in the Environment contributes about 279/900 ≈ 31%, while the initialization stage accounts for only about 10% of the total time.
>
> In terms of dollar cost, the initialization stage only requires the LLM to propose suitable model combinations, resulting in a very small cost, about 10% of the total. The remaining cost is incurred during the iterative search stage.
>
> |**Field(Unit)**|**Value**|**Percentage**|
> |---------------------|---------|--------------|
> |agent_total_time(s)|900.31|100.00%|
> |initial_time(s)|94.23|10.46%|
> |trial_generation_time(s)|526.71|58.49%|
> |eval_time(s)|279.37|31.03%|
> |total_dollar(USD)|0.12|100.00%|
> |initial_dollar(USD)|0.01|10.08%|
> |search_dollar(USD)|0.11|89.92%|
>
> > what if we don't pre-assign roles?
>
> We provide an ablation study without assigning roles in the paper. This result is shown in the last row of Table 2. For your convenience, we extract them below:
>
> Method | MATH | MMLU |
> |----|----|----|
> Ours | 56 | 54 |
> Ours w/o Role | 52 | 54 |
>
> We perform an ablation by removing role setting, letting all nodes process predecessors’ outputs and generate new answers, which degrades graph performance on MATH. This highlights the importance of the fuser–assistant role division in test-time scaling. We note that MMLU performance remains stable without Insight~3 or role settings, as it favors larger models with fewer nodes, reducing the impact of roles.
>
> We hope our replies clarify your concerns. Let us know if you have further questions.

---

> > ### Comment · Reviewer_yy82 · 2025-11-27
> >
> > thank you for your reply.

---

> > > ### Author Response · Authors · 2025-11-27
> > >
> > > We thank the reviewer for the valuable comments again. Please let us know if any concerns remain. If our responses have satisfactorily addressed all issues, we would appreciate your consideration of an increased rating score.

---

### Official Review · Reviewer_mctA · 2025-10-31

**Soundness:** 2
**Presentation:** 2
**Contribution:** 2
**Rating:** 4
**Confidence:** 4

**Summary:**

This paper addresses the problem of finding compute-optimal multi-LLM collaboration architectures for test-time scaling (TTS). The authors formulate TTS as a multi-LLM collaboration graph where nodes represent LLMs with assigned roles (fuser/assistant) and edges denote information flow. They identify three empirical insights about TTS behavior: (1) tasks prefer specific model family/size combinations, (2) width and depth have task-dependent optima, and (3) width-depth interdependence. Based on these insights, they propose Agent-REINFORCE, an LLM-agent-augmented framework that mirrors REINFORCE's sample-gradient-update pipeline using textual feedback as gradients. Experiments on MATH, MMLU, and HumanEval demonstrate superior search efficiency and performance compared to traditional and LLM-based baselines.

**Strengths:**

- Well-motivated problem: The paper identifies a real gap in existing TTS methods—fixed architectures and single-model usage—and proposes a principled solution.
- Comprehensive empirical analysis: The pilot experiments (Section 4) provide valuable insights that generalize across tasks and inform the algorithm design.
- Strong experimental results: Agent-REINFORCE consistently outperforms baselines in both search efficiency (convergence time) and final performance across all three benchmarks.
- Practical framework: The method handles real-world constraints (budget, latency) and supports multiple cost metrics (FLOPs, wall-clock time, monetary cost).

**Weaknesses:**

- Limited scope of evaluation: Only three tasks (MATH, MMLU, HumanEval) from limited domains (math, knowledge, code). Only LLaMA and Gemma models tested; unclear how insights generalize to other model families. No evaluation on truly large-scale models (e.g., 70B+) due to computational constraints.
- Computational cost of search: The paper does not report the cost of running DeepSeek-R1 as the search agent. Search time includes only environment evaluation, not LLM agent inference. For practical deployment, the amortization of search cost needs analysis.
- Insight generality: Insight 1 (model family preference) may be dataset/domain-specific. Insight 2 (non-monotonic scaling) is known in TTS literature but not cited adequately. Insight 3 (width-depth interdependence) is intuitive; the novelty is primarily empirical validation
- Statistical rigor: Limited discussion of variance across runs. Some results show confidence intervals (Fig. 3c-d) but most do not. No significance testing for performance differences.

**Questions:**

- Search cost amortization: How many test queries are needed to amortize the search cost? Can you provide a break-even analysis comparing search time vs. inference time savings?
- Insight generalization: How do the three insights hold for tasks outside your evaluation set (e.g., long-form generation, multimodal tasks, code debugging)? Have you tested on additional domains?
- Agent model choice: Why DeepSeek-R1? How sensitive are results to the choice of search agent? Have you tried smaller/larger agents or different model families?
- Graph constraints: The current formulation allows arbitrary DAGs. Would restricting to specific topologies (trees, chains) improve efficiency while maintaining performance?
- Budget sensitivity: How does performance change with budget granularity? The paper tests specific budgets (18, 42, 80) but not the full spectrum.
- Multi-objective optimization: The latency-aware optimization is mentioned briefly. Can you provide more details on the Pareto front and trade-off curves?
- Online adaptation: Can the searched graph adapt during deployment if data distribution shifts? Or is it fixed post-search?

---

> ### Author Response · Authors · 2025-11-26
> **Response to Reviewer mctA: Part I**
>
> We appreciate the reviewer’s valuable comments. Below, we respond to each point in detail. (For your convenience in reviewing the rebuttal sequentially, we have removed the earlier partial responses and reorganized the replies without changing previous content.)
>
> > W1 Limited scope of evaluation: Only three tasks (MATH, MMLU, HumanEval) from limited domains (math, knowledge, code). Only LLaMA and Gemma models tested; unclear how insights generalize to other model families. No evaluation on truly large-scale models (e.g., 70B+) due to computational constraints.
>
> > Q2 Insight generalization: How do the three insights hold for tasks outside your evaluation set?
>
> Because W1 and Q2 correspond to the same concern, we address them together in a unified response.
>
> Firstly, for the limited scope of evaluation and no evaluation on the 70B+ model, we use the MedQA dataset to evaluate whether our method can generalize to the medical domain, and we select LLaMA-3.1-70B as a representative model in the 70B+ scale. The budget is set to 274 units of FLOPs, which corresponds to the computational cost of using a single 70B model. As shown in the table below, our approach is still able to search for a higher-performing TTS graph while requiring substantially less search time and achieving lower inference latency. These results demonstrate that our method generalizes effectively across domains and model sizes.
>
> | Method   | Acc  | Search Time | Inference Time |
> | -------- | ---- | ----------- | -------------- |
> | BO       | 34   | 3098        | 65.43          |
> | TextGrad | 37   | 3844        | 24.70          |
> | Ours     | 54   | 537         | 21.37          |
>
> Secondly, for the concern about how insights generalize to other model families, to evaluate the generalizability of our insights across different model families and domains, we conduct experiments using Qwen-2.5 on the MedQA dataset. The results are summarized as follows. Table 1 shows that repeating the preferred model family on MedQA yields better performance; Table 2 demonstrates that MedQA prefers a single 7B Qwen-2.5 model, likely due to the dataset’s reliance on domain-specific medical knowledge, which favors larger models. These observations support the generalizability of Insight 1. Table 3 reports the scaling behavior under different graph widths and depths using Qwen-2.5-1.5B. The results show that performance increases and then decreases as the width or depth grows, confirming the generalizability of Insight 2. Moreover, as width (or depth) increases, the optimal depth (or width) shifts downward, illustrating the generalizability of Insight 3.
>
> Table 1: Model family preference on MedQA with Qwen-2.5 and Gemma.
> | Model Combination | MedQA |
> |-------------------|-------|
> | Gemma-1B             | 11    |
> | Qwen-1.5B              | 26    |
> | Qwen-1.5B$\times 1$ + Gemma-1B$\times 1$  | 19    |
> | Qwen-1.5B$\times 2$         | 29    |
> | Gemma-1B$\times 2$           | 13    |
>
> Table 2: Model size preference on MedQA with Qwen-2.5.
> | Model Combination (Qwen-2.5)       | Acc |
> |-------------------------|-----|
> | 9 × 1.5B                | 28  |
> | 3 × 3B                  | 45  |
> | 2 × 3B + 3 × 1.5B       | 34  |
> | 1 × 7B                  | 48  |
>
> Table 3: Performance under various Width-Depth collaboration graphs on MedQA. Model is Qwen-2.5-1.5B.
>
> |      | 1  | 2  | 4  | 8  | 12 | 14 |
> |-------------------|----|----|----|----|----|----|
> | Depth:1  | 28 | 28 | 29 | 29 | 27 | 28 |
> | Depth:2  | 31 | 31 | 33 | 29 | —  | —  |
> | Depth:3  | 35 | 33 | 31 | —  | —  | —  |
> | Width:1     | 28 | 29 | 30 | 34 | 35 | 28 |
> | Width:2     | 34 | 36 | 32 | 30 | —  | —  |
> | Width:3     | 34 | 29 | 31 | —  | —  | —  |

---

> ### Author Response · Authors · 2025-11-26
> **Response to Reviewer mctA: Part II**
>
> > W2 Computational cost of search: The paper does not report the cost of running DeepSeek-R1 as the search agent. Search time includes only environment evaluation, not LLM agent inference. For practical deployment, the amortization of search cost needs analysis.
>
> > Q1 Search cost amortization: How many test queries are needed to amortize the search cost? Can you provide a break-even analysis comparing search time vs. inference time savings?
>
> Because W2 and Q1 correspond to the same concern, we address them together in a unified response.
>
> Firstly, for the search cost, we report the inference cost (both time and dollar cost) on the MATH dataset under a budget of 80, separating the initial stage and the search iteration stage, as in the following table. The overall time and dollar cost of our method remain small because the training set we use is small.
>
> In terms of time cost, the search stage involves up to 30 iterations, corresponding to 30 batches, and is divided into two parts: trial generation by the LLM agent and trial execution in the Environment. The trial-generation stage dominates the total search time, accounting for approximately 526/900 ≈ 58% of the total. Execution in the Environment contributes about 279/900 ≈ 31%, while the initialization stage accounts for only about 10% of the total time.
>
> In terms of dollar cost, the initialization stage only requires the LLM to propose suitable model combinations, resulting in a very small cost, about 10% of the total. The remaining cost is incurred during the iterative search stage.
>
> | **Field (Unit)**      | **Value** | **Percentage** |
> | --------------------- | --------- | -------------- |
> | agent_total_time (s) | 900.31    | 100.00%        |
> | initial_time (s)     | 94.23     | 10.46%         |
> | trial_generation_time (s)      | 526.71    | 58.49%         |
> | eval_time (s)        | 279.37    | 31.03%         |
> | total_dollar (USD)   | 0.12      | 100.00%        |
> | initial_dollar (USD) | 0.01      | 10.08%         |
> | search_dollar (USD)  | 0.11      | 89.92%         |
>
> Secondly, for amortizing search cost, we consider both time and monetary expenditure.
>
> Time Cost. As shown in Tables 1 and 3 of the main paper, our method achieves higher accuracy, lower search time, and lower inference latency than the baselines, so amortization is unnecessary from a time perspective. The primary reason for our faster and more effective search is that our method recognizes that larger budgets do not always yield better performance. Consequently, during search, it often gravitates toward smaller, high-performing computation graphs, resulting in improved accuracy and speed.
>
> Dollar Cost. Our search dollar cost is higher than Random, BO, and GPTSwarm because these baselines do not employ a search agent during optimization and therefore need to amortize their inference cost. For methods such as TextGrad, both the search cost and inference cost are substantially larger, so amortization is unnecessary.
> Taking BO as an example: its search cost on MATH is \\$0.017, whereas ours is \\$0.121. The inference cost for BO is $\\$4.8\times 10^{-4}$ per query, compared to $\\$3.1\times 10^{-4}$ for ours. Thus, the amortization threshold is
> $$
> \frac{0.121 - 0.017}{4.8\times 10^{-4} - 3.1\times 10^{-4} } = 612.
> $$
> This means that after approximately 612 test queries, our method becomes more cost-effective than BO when amortized over repeated test-time usage. The detailed search and inference costs are shown in the table below. Using the same calculation, the amortization thresholds for Random and GPTSwarm are 439 and 785, respectively. These values are not large in practical deployments, where typical usage quickly exceeds these thresholds. We also emphasize that although our search incurs a higher dollar cost, it yields substantially better performance, making the additional cost well justified.
>
> Table: Search Cost and Inference Cost of Various Methods. Search Cost of Random, BO, GPTSwarm is from their TTS execution on the training set.
> | Method       | Search Cost (USD)| Inference Cost ($\times 10^{-4}$USD per query) |
> |--------------|--------------|------------------------|
> | Random  | 0.020        | 5.4                  |
> | BO           | 0.017        | 4.8                  |
> | GPTSwarm     | 0.011        | 4.5                  |
> | TextGrad     | 0.188        | 4.6                  |
> | Ours         | 0.121        | 3.1                  |

---

> ### Author Response · Authors · 2025-11-26
> **Response to Reviewer mctA: Part III**
>
> > W3 Insight generality: Insight 1 (model family preference) may be dataset/domain-specific. Insight 2 (non-monotonic scaling) is known in TTS literature but not cited adequately. Insight 3 (width-depth interdependence) is intuitive; the novelty is primarily empirical validation.
>
> We address the concerns for the three insights.
>
> **1.** For Insight 1, you are correct that model family preference is both domain-specific and task-specific. This is precisely what we aim to highlight, and it underscores the importance of selecting an appropriate model family during search initialization.
>
> **2.** For insight 2, we have added the relevant citations in the main text. Specifically, we add [1,2,3,4].
>
> [1] Wang, Jian, et al. "Scaling over Scaling: Exploring Test-Time Scaling Pareto in Large Reasoning Models." arXiv preprint arXiv:2505.20522 (2025).
>
> [2] Tang, Yung-Chen, Pin-Yu Chen, and Andrea Cavallaro. "CarBoN: Calibrated Best-of-N Sampling Improves Test-time Reasoning." arXiv preprint arXiv:2510.15674 (2025).
>
> [3] Brown, Bradley, et al. "Large language monkeys: Scaling inference compute with repeated sampling." arXiv preprint arXiv:2407.21787 (2024).
>
> [4] Li, Junyou, et al. "More agents is all you need." arXiv preprint arXiv:2402.05120 (2024).
>
> **3.** For Insight 3, we provide a theoretical analysis:
>
>
> part 1. Depth $d$: refinement as a Markov process
>
> Let $p_o$: initial correctness probability, $p_{r1}$: probability refinement fixes an incorrect answer, and $p_{r2}$: probability refinement corrupts a correct answer.
> Let $p_t$ be correctness after $t$ refinement steps:
> $$
> p_{t+1}
> = p_t(1 - p_{r2}) + (1 - p_t)p_{r1}
> = p_t(1 - p_{r1} - p_{r2}) + p_{r1}.
> $$
>
> Closed form:
>
> $$
> p_t = p^\* + (p_o - p^\*)(1 - p_{r1} - p_{r2})^t,
> \qquad
> p^\* = \frac{p_{r1}}{p_{r1} + p_{r2}}.
> $$
>
> We could conclude: (1) If $p_o < p^\*$: accuracy increases, then saturates. (2) If $p_o > p^\*$: accuracy decreases toward $p^\*$.  (3) Depth $d$ has an optimal value before saturation or decline.
>
> Part 2. Width $k$: fusion with capacity limits
>
> Let $p_d$ be the correctness after depth $d$. Fusion of $k$ samples yields $p_f(k, p_d)$.
>
> Ideal fusion:
>
> $$
> p_f^{\text{ideal}}(k, p_d) = 1 - (1 - p_d)^k.
> $$
>
> With long-context degradation:
>
> Let $g(k)$ be a penalty:
>
> $$
> g(k) = 1 - \beta \max(0, k - k_0).
> $$
>
> Final fusion correctness:
>
> $$
> p_f(k, p_d) = g(k)\,\big[1 - (1 - p_d)^k\big].
> $$
>
> Implications: (1) Small $k$: accuracy increases with width.  (2) Large $k$: penalty dominates $\Rightarrow$ accuracy saturates or declines.  (3) Width $k$ has an optimal value.
>
> part 3. Interdependence between width $k$ and depth $d$
>
> Since:
>
> $$
> p_f(k, d) = p_f\big(k, p_d(p_o, p_{r1}, p_{r2})\big),
> $$
>
> The optimal width depends on depth:
>
> $$
> k^\*(d) = \arg\max_k p_f(k, d).
> $$
>
> Similarly, the optimal depth depends on the width:
>
> $$
> d^\*(k) = \arg\max_d p_f(k, d).
> $$
>
> Implications: (1) When $d$ is small $\Rightarrow p_d$ is low $\Rightarrow$ larger $k$ is beneficial.  (2) When $d$ is large $\Rightarrow$ marginal gain shrinks and long-context penalty appears earlier $\Rightarrow$ optimal $k$ is smaller.  (3) Likewise, strong width reduces the required depth.
>
> > W4 Statistical rigor: Limited discussion of variance across runs. Some results show confidence intervals (Fig. 3c-d) but most do not. No significance testing for performance differences.
>
> We provide significance testing results for BO, TextGrad, and our method on MMLU dataset below (due to time constraints, only partial results are included; the full results will be added in the final revision). Although the absolute accuracies vary slightly, the relative ranking remains unchanged, further demonstrating the superior performance of our method.
>
> | Method   | Acc            |
> | -------- | -------------- |
> | BO       | 36.8 $\pm$ 5.2 |
> | TextGrad | 39.9 $\pm$ 3.6 |
> | Ours     | 61.5 $\pm$ 5.1 |
>
> > Q3 Agent model choice: Why DeepSeek-R1? How sensitive are results to the choice of search agent? Have you tried smaller/larger agents or different model families?
>
> We experimented with multiple LLMs as the search agent, including DeepSeek-V3, Doubao-1.5-Pro, and LLaMA-3-70B, on the MATH dataset under a budget of 80. The results are shown below:
>
> | Model          | MATH (budget=80) |
> | -------------- | ---------------- |
> | DeepSeek-R1    | 56               |
> | DeepSeek-V3    | 50               |
> | LLaMA-3-70B   | —                |
> | DouBao-1.5-Pro | 51               |
>
> LLaMA-3-70B failed because it could not reliably follow instructions and therefore did not produce valid step-wise outputs. In contrast, our method remains effective when applied to models with strong reasoning capabilities. DeepSeek-R1 achieves the best performance, suggesting that our approach benefits from a search agent with strong instruction-following and reasoning abilities.

---

> ### Author Response · Authors · 2025-11-26
> **Response to Reviewer mctA: Part IV**
>
> > Q4 Graph constraints: The current formulation allows arbitrary DAGs. Would restricting to specific topologies (trees, chains) improve efficiency while maintaining performance?
>
> We evaluate both chain and tree topologies, and the results on MATH with a budget of 80 are shown below. The optimal performance achieved by either chain or tree is substantially lower than that of our graph topology. This indicates that the graph structure is essential for configuring effective test-time scaling.
>
> | Topology | Acc |
> |----------|----------|
> | Chain    | 47                 |
> | Tree     | 49                 |
> | Ours     | 56                 |
>
> > Q5 Budget sensitivity: How does performance change with budget granularity? The paper tests specific budgets (18, 42, 80) but not the full spectrum.
>
>
> We provide additional results using finer-grained budgets on the MATH dataset. The performance across budget ranges is shown below:
>
> | budget range (on MATH) | ACC |
> | ------------------- | ----- |
> | 1–10                | 33    |
> | 10–18               | 40     |
> | 18–29               | 47    |
> | 29–51               | 50    |
> | 51–80               | 56    |
> | >80                 | 56    |
>
> These results show a clear trend: performance improves steadily as more budget becomes available and then stabilizes once the budget exceeds the optimal range. After this point, allocating additional compute does not yield further performance gains, as the search has already reached its effective budget. This supports our claim that our method is budget-sensitive and exhibits saturation behavior beyond the optimal compute level.
>
>
> > Q6 Multi-objective optimization: The latency-aware optimization is mentioned briefly. Can you provide more details on the Pareto front and trade-off curves?
>
> We provide additional details in Appendix A.9 in the paper. Specifically, our method supports multi-objective optimization by incorporating latency into the feedback signal, enabling the search agent to recognize which graph structures are more time-efficient. We report the corresponding trade-off results below:
>
> Table. Ours is our method with only a single performance objective, while Ours (joint) is our method with a performance and latency multi-objective.
> | Budget setting | Method       | Acc | Latency (Inf.) |
> | -------------- | ------------ | --- | -------------- |
> | Dollar ≤ 5E-4   | Ours         | 56  | 18.1           |
> |                | Ours (joint) | 52  | **14.0**       |
> | FLOPs Budget 42       | Ours         | 50  | 23.8           |
> |                | Ours (joint) | 47  | **17.4**       |
> | FLOPs Budget 18       | Ours         | 47  | 16.7           |
> |                | Ours (joint) | 44  | **5.6**        |
>
> These results show that when latency is not considered, our method achieves higher performance. When latency is included as an additional optimization objective, accuracy decreases slightly (by 3–4 points), but latency is significantly reduced (by 22–66%). The two configurations represent different points along the Pareto frontier, illustrating a meaningful trade-off between accuracy and latency. This demonstrates that our approach can flexibly adapt its search behavior to match the user’s optimization priorities.
>
>
> > Q7 Online adaptation: Can the searched graph adapt during deployment if data distribution shifts? Or is it fixed post-search?
>
> Our current method is designed for an offline setting, where the search is performed on a fixed training set and the resulting graph is applied directly during testing. Therefore, we do not explicitly address distribution shift in the proposed method. We agree that distribution shift is an important and challenging problem. To extend our framework to the online setting, where samples arrive sequentially, and no ground-truth labels are available, one can incorporate external evaluators such as LLM-as-a-Judge and perform real-time test-time scaling–based probabilistic graph search. This would allow the system to continuously update its graph structure in response to changing data distributions. Because the search cost of our method is relatively low, we believe it is well-suited for such adaptive online scenarios.
>
>
> We hope our replies clarify your concerns. Let us know if you have further questions.

---

> ### Author Response · Authors · 2025-11-27
>
> Dear Reviewer mctA,
>
> Thank you again for your thoughtful and comprehensive comments. We have now provided complete responses to all of your questions. We look forward to hearing your feedback and would be glad to further clarify any remaining concerns.

---

### Official Review · Reviewer_GAUb · 2025-11-01

**Soundness:** 2
**Presentation:** 3
**Contribution:** 3
**Rating:** 6
**Confidence:** 4

**Summary:**

This paper addresses an important yet underexplored problem in test-time scaling (TTS): searching for compute-optimal multi-LLM collaboration graphs under a fixed budget for specific tasks. The authors formalize this as a multi-LLM collaboration graph optimization problem, where nodes represent LLM primitives with role assignments (fuser/assistant) and model assignments, and edges represent information flow. Through pilot experiments, the authors identify three key insights: task-specific preferences for model combinations, task-dependent optima for width/depth beyond which additional compute yields negative returns, and interdependence between graph width and depth. Based on these insights, they propose Agent-REINFORCE, which maps REINFORCE's sample-gradient-update pipeline to sample-feedback-update, leveraging an LLM agent to incorporate prior knowledge for efficient search. Experiments demonstrate that the method significantly outperforms both traditional methods and LLM-based baselines on MATH, MMLU, and HumanEval datasets in terms of both accuracy and search efficiency, while effectively handling joint optimization objectives for accuracy and latency.

**Strengths:**

1. Novel and important problem formulation: This work is the first to systematically formalize test-time scaling as a multi-LLM collaboration graph optimization problem, filling a critical gap in existing work. While most TTS methods consider only fixed topologies and single models, this paper jointly optimizes architecture, role assignments, and heterogeneous model combinations, offering significant theoretical and practical value to the multi-agent collaboration community.

2. Deep and actionable empirical insights: The three insights discovered through systematic pilot experiments provide both strong algorithmic guidance and valuable empirical understanding of TTS behavior. These insights are thoroughly validated in ablation studies, demonstrating their crucial role in accelerating convergence and improving performance.

3. Clever method design with solid execution: The Agent-REINFORCE framework innovatively replaces REINFORCE's gradient updates with LLM-generated textual feedback, effectively combining the data efficiency of gradient methods with the semantic understanding of LLMs. The multi-stage initialization strategy is particularly clever in significantly reducing the search space, and the experimental design is comprehensive across multiple datasets and optimization scenarios.

4. Convincing experimental results: The method comprehensively outperforms multiple baselines across three representative tasks in both accuracy and search efficiency. Ablation studies clearly demonstrate the contribution of each component, and the method shows good generalization across different budget metrics and joint optimization objectives.

**Weaknesses:**

1. Incomplete baseline comparisons: While the paper compares against GPTSwarm and TextGrad, it misses important baselines in agent workflow optimization such as AFlow and ADAS, which have become standard comparison methods in this field. Including experimental comparisons and detailed discussions with these methods would better position the work and demonstrate its advantages.
2. The paper lacks quantification of the inference cost of the agent itself during the search process. It would be valuable to explicitly report the agent's FLOPs/dollar overhead and its proportion in the total search budget to comprehensively assess the method's practical efficiency.
3. The experiments use dated benchmarks for TTS research. Evaluating on more recent benchmarks such as LiveCodeBench would strengthen the contributions. Additionally, expanding the model families beyond LLaMA and Gemma to include more widely-used backbones like Qwen would enhance the generalization claims.

**Questions:**

As shown in weakness

---

> ### Author Response · Authors · 2025-11-21
> **Part I Response to Reviewer GAUb**
>
> We sincerely thank the reviewer for the helpful suggestions and guidance on the search cost and evaluation suggestions. We reply to your concerns one by one.
>
> > W2 The paper lacks quantification of the inference cost of the agent itself during the search process. It would be valuable to explicitly report the agent's FLOPs/dollar overhead and its proportion in the total search budget to comprehensively assess the method's practical efficiency.
>
>
> We report the inference cost (both time and dollar cost) on the MATH dataset under a budget of 80, separating the initial stage and the search iteration stage, as in the following table. The overall time and dollar cost of our method remain small because the training set we use is small.
>
> In terms of time cost, the search stage involves up to 30 iterations, corresponding to 30 batches, and is divided into two parts: trial generation by the LLM agent and trial execution in the Environment. The trial-generation stage dominates the total search time, accounting for approximately 526/900 ≈ 58% of the total. Execution in the Environment contributes about 279/900 ≈ 31%, while the initialization stage accounts for only about 10% of the total time.
>
> In terms of dollar cost, the initialization stage only requires the LLM to propose suitable model combinations, resulting in a very small cost, about 10% of the total. The remaining cost is incurred during the iterative search stage.
>
> | **Field (Unit)**      | **Value** | **Percentage** |
> | --------------------- | --------- | -------------- |
> | agent_total_time (s) | 900.31    | 100.00%        |
> | initial_time (s)     | 94.23     | 10.46%         |
> | trial_generation_time (s)      | 526.71    | 58.49%         |
> | eval_time (s)        | 279.37    | 31.03%         |
> | total_dollar (USD)   | 0.12      | 100.00%        |
> | initial_dollar (USD) | 0.01      | 10.08%         |
> | search_dollar (USD)  | 0.11      | 89.92%         |

---

> ### Author Response · Authors · 2025-11-27
> **Part II Response to Reviewer GAUb**
>
> > W1 Incomplete baseline comparisons: ... it misses important baselines in agent workflow optimization such as AFlow and ADAS, ...
>
> > W3 Evaluating on more recent benchmarks such as LiveCodeBench would strengthen the contributions. Additionally, expanding the model families beyond LLaMA and Gemma to include more widely-used backbones like Qwen would enhance the generalization claims.
>
> To address the reviewer’s concerns more efficiently, we conducted additional experiments on LiveCodeBench using the additional Qwen model and incorporating two new baselines, ADAS and AFlow. For this reason, we address W1 and W3 together.
>
> Following the reviewer's instruction, we additionally conduct experiments using the Qwen-2.5-[3B, 7B] model on the LiveCodeBench dataset under an FLOPs budget of 80. Two new baselines are included: AFlow and ADAS. Both AFlow and ADAS implement agent workflows through code. To adapt them to our task, we provide ADAS with basic model components and supply AFlow with fundamental parallel and sequential operations. These methods use a Meta Agent as the optimizer to iteratively refine combinations of these primitive modules. The experimental results are shown in the table below.
>
> | Method       | Pass@1 | Search Time (s) | Inference Time (s) |
> |--------------|--------------|-------------|----------------|
> | BO           | 18           | 1188        | 20.1           |
> | TextGrad     | 19           | 1788        | 22.1           |
> | GPTSwarm     | 8            | 720         | 26.89          |
> | Aflow        | 12           | 552         | 14.37          |
> | ADAS         | 8            | 611         | 16.95          |
> | Ours         | 28           | 273         | 5.83           |
>
> Our analysis of the results is as follows: (1) Our method continues to achieve the best performance on LiveCodeBench (higher test-set pass@1, faster search, and lower inference latency), demonstrating strong generalization. (2) Our method outperforms both ADAS and AFlow. This is primarily because ADAS and AFlow are designed for general workflow construction and do not account for the characteristics of test-time scaling. Benefiting from our proposed insights, our method achieves substantially superior performance.
>
>
>
> Thank you again for your thoughtful review. We welcome any additional feedback.

---

### Author Response · Authors · 2025-12-03

Dear Area Chair,

We appreciate the AC’s careful evaluation of our submission. To support the assessment process of the new AC, we offer a summary that outlines the strengths from reviewers and essential points discussed and clarified during the rebuttal.

## Summary of Strengths

### **1. Novel and Well-Motivated Problem Formulation**

- First to formalize **test-time scaling as a multi-LLM collaboration graph optimization problem**, addressing a clear gap in existing TTS methods (*from Reviewers GAUb, mctA, UoVT*).
- Moves beyond fixed architectures and single-model usage by **jointly optimizing collaboration topology, role assignment, and heterogeneous model combinations** (*from Reviewers GAUb, mctA*).
- The direction of **multi-LLM collaboration (on TTS)** is timely and important (*from Reviewer UoVT*).


### **2. Strong Empirical Insights and Analysis**

- Section 4’s pilot experiments provide **deep, actionable empirical insights** that generalize across tasks (*from Reviewers GAUb, mctA*).
- Insights are shown to directly guide algorithm design and improve search efficiency, supported by **clear ablation studies** (*from Reviewers GAUb, yy82*).


### **3. Effective and Innovative Method Design**

- Agent-REINFORCE offers a novel mechanism by using **LLM-generated textual feedback instead of gradient updates**, combining semantic reasoning with data efficiency (*from Reviewer GAUb*).
- **Multi-stage initialization** effectively reduces the search space and accelerates convergence (*from Reviewer GAUb*).
- The framework supports **multiple cost metrics** (FLOPs, monetary cost) and handles real-world constraints (*from Reviewers mctA, yy82*).

### **4. Comprehensive and Convincing Experimental Results**

- Demonstrates consistent **superiority over baselines** in accuracy and search efficiency across three benchmarks (*from Reviewers GAUb, mctA, yy82*).
- Includes detailed **ablation studies**, justifications, and analyses showing the contribution of each component (*from Reviewers GAUb, yy82*).
- Results hold across **different budget formulations and joint optimization objectives** (*from Reviewers GAUb, yy82*).

### **5. Clear and Well-presented Writing**

- The design is clearly explained, with strong writing quality and informative figures (*from Reviewer yy82*).


## **Essential Points Clarified During the Rebuttal**

### 1. **Search Cost Quantification & Amortization**

*Reviewers GAUb, mctA, yy82* requested quantitative analysis of the agent’s search cost. We reported detailed **time cost**, **dollar cost**, and **amortization analysis**. Results show that the overall inference and search cost is small; amortization requires only **612 queries** for dollar cost, and our method outperforms baselines in both search time and inference efficiency.
### 2. **Additional Benchmarks, Models, and Baselines**

In response to broader evaluation requests (*Reviewers GAUb, mctA*), we added experiments on **Qwen-2.5 3B/7B**, **LiveCodeBench**, and **MedQA**, and incorporated new baselines **AFlow** and **ADAS**. Results confirm strong **generalization** across models, datasets, and optimization settings.
### 3. **Generalization Across Architectures, Topologies, and Backbones**

*Reviewer mctA* requested experiments on different topologies, larger models (70B+), and alternative agent backbones. We added corresponding results showing robustness under all these settings, including budget sensitivity.
### 4. **Clarifications on Methodology and Definitions**

*Reviewer mctA* requested clarification on distribution shifts and latency–performance Pareto fronts; we provided detailed explanations. *Reviewer UoVT* pointed out terminology issues (e.g., “Theorem”), which we corrected.
### 5. **Role Assignment and Extension to New Roles**

For concerns regarding role assignment (*Reviewers yy82, UoVT*), we added ablations without pre-assigned roles, showing task-dependent importance of roles. We also discussed how the framework generalizes to additional role types.
### 6. **Scaling Behavior**

*Reviewer UoVT* requested scaling curves. We provided scaling results showing performance increases and eventual saturation/decrease as the budget grows.

We hope this summary supports the Area Chair in efficiently and accurately assessing our submission.

Thanks,
All Authors

---

### Meta-Review · Area_Chair_L6fz · 2025-12-16

**Summary:**

This paper formulates Test-Time Scaling (TTS) as a search problem on a multi-LLM cooperative graph consisting of "roles (assistant/fuser) × model assignment × information flow." It proposes Agent-REINFORCE, a search framework that updates a stochastic graph distribution using text feedback from LLM agents. The study demonstrates that on MATH, MMLU, and HumanEval, the proposed method can find high-performance and low-latency configurations with fewer searches compared to random search, Bayesian Optimization (BO), and existing gradient/LLM-based search methods (GPTSwarm, MaaO, TextGrad, etc.).

In the author response, the following were presented, increasing the practicality and clarity of the paper's positioning: (i) the addition of single-model baselines; (ii) a breakdown of search agent time/monetary costs and an amortized analysis; (iii) follow-up experiments on additional tasks/models (e.g., LiveCodeBench + Qwen, MedQA + 70B class models); and (iv) supplements regarding roles, theorem notation, qualitative examples, and scaling plots.

**Reviewer Concerns:**

* **Validity of Comparisons (Single Model / Additional Baselines)** (yy82, GAUb): There was a lack of differentiation from single models and comparisons with agent workflow methods (such as AFlow/ADAS). The authors presented a table for a single-model (Llama3-8B) baseline and further reported additional comparisons including AFlow/ADAS on LiveCodeBench (using Qwen-2.5).
* **Search Cost (Agent Inference Overhead) and Amortization** (GAUb, mctA, yy82): Previous "search time" metrics focused on environment evaluation, leaving the cost of the search agent (DeepSeek-R1) unclear. The authors presented a breakdown of time and cost into agent inference, environment evaluation, and initialization, as well as an estimate of amortization thresholds against BO and other methods.
* **Generality/Scope (Tasks, Model Families, Large Models)** (mctA, GAUb): Concerns were raised regarding the bias toward three specific tasks and limited models. The authors presented additional results on MedQA, LiveCodeBench, and Qwen (along with examples using 70B-class models), reinforcing this aspect to a certain extent.
* **Rigor of Description/Positioning** (mctA): The relationship with existing research (citations) in Insight 2 and the handling of statistical robustness (variance/significance) are weak. While there are supplements such as updated results with multiple runs in some parts, there remains room for improvement overall.
* **Role Design, Theorem Notation, Qualitative Examples, Scaling Plots** (UoVT, yy82): Issues included roles being manually fixed, the validity of "Theorem" notation, a lack of qualitative behavioral examples, and insufficient scaling visualization. The authors presented (1) directions for additional roles/flexibility, (2) a correction from "Theorem" to "Proposition," (3) the addition of qualitative examples, and (4) a table showing performance trends as the budget increases.

**Reviewer Scores:**

* **Reviewer GAUb** (Initial: 6): Major concerns were largely resolved by the presentation of additional baselines (AFlow/ADAS), additional tasks (LiveCodeBench), and the agent cost breakdown. **Maintains 6 → 6 (High Confidence)**.
* **Reviewer mctA** (Initial: 4): Scope expansion (MedQA/70B, Qwen, etc.), amortization/costs, and detailing of multi-objective optimization mitigated many concerns. However, the rigor of citations and statistics remains weak, and a decisive factor to move the score is uncertain. **Maintains 4 → 4 (Medium to High Confidence; possibility of 4$\rightarrow$6, but certainty is not high)**.
* **Reviewer yy82** (Initial: 6): Concerns were largely resolved by the single-model comparison, agent cost breakdown, and role ablation. **Maintains 6 → 6 (High Confidence)**.
* **Reviewer UoVT** (Initial: 6): Concerns were largely resolved by the discussion on role expansion/flexibility, Theorem notation correction, qualitative examples, and scaling supplements. **Maintains 6 → 6 (High Confidence)**.

**Predicted Average:** (6 + 4 + 6 + 6) / 4 = **5.5**

---

### Decision · Program_Chairs · 2026-01-26

Reject